# Diversity in Ruby Geochemistry and Its Inclusions: Intra- and Inter- Continental Comparisons from Myanmar and Eastern Australia

**Frederick L. Sutherland** [1,*], **Khin Zaw** [2], **Sebastien Meffre** [2], **Jay Thompson** [2], **Karsten Goemann** [3], **Kyaw Thu** [4], **Than Than Nu** [5], **Mazlinfalina Mohd Zin** [6] and **Stephen J. Harris** [7]

1   Mineralogy & Petrology, Geosciences, Australian Museum, 1 William Street, Sydney, NSW 2010, Australia
2   CODES Centre of Ore Deposit and Earth Sciences, University of Tasmania, Hobart, Tas 7001, Australia; khin.zaw@utas.edu.au (K.Z.); sebastien.meffre@utas.edu.au (S.M.); jay.thompson@utas.edu.au (J.T.)
3   Central Science Laboratory, University of Tasmania, Hobart, Tas 7001, Australia; karsten.goemann@utas.edu.au
4   Geology Department, Yangon University, Yangon 11041, Myanmar; macle45@gmail.com
5   Geology Department, Mandalay University, Mandalay 05032, Myanmar; thanthannu83@gmail.com
6   Geology Programme, Faculty of Science and Technology, The National University of Malaysia (UKM), Selangor 43600, Malaysia; farlinnzin@gmail.com
7   School of Biological, Earth and Environmental Sciences, University of New South Wales, Sydney, NSW 2052, Australia; s.j.harris@student.unsw.edu.au
*   Correspondence: linsutherland1@gmail.com; Tel.: +61-2-65826553

**Abstract:** Ruby in diverse geological settings leaves petrogenetic clues, in its zoning, inclusions, trace elements and oxygen isotope values. Rock-hosted and isolated crystals are compared from Myanmar, SE Asia, and New South Wales, East Australia. Myanmar ruby typifies metasomatized and metamorphic settings, while East Australian ruby xenocrysts are derived from basalts that tapped underlying fold belts. The respective suites include homogeneous ruby; bi-colored inner (violet blue) and outer (red) zoned ruby; ruby-sapphirine-spinel composites; pink to red grains and multi-zoned crystals of red-pink-white-violet (core to rim). Ruby ages were determined by using U-Pb isotopes in titanite inclusions (Thurein Taung; 32.4 Ma) and zircon inclusions (Mong Hsu; 23.9 Ma) and basalt dating in NSW, >60–40 Ma. Trace element oxide plots suggest marble sources for Thurein Taung and Mong Hsu ruby and ultramafic-mafic sources for Mong Hsu (dark cores). NSW rubies suggest metasomatic (Barrington Tops), ultramafic to mafic (Macquarie River) and metasomatic-magmatic (New England) sources. A previous study showed that Cr/Ga vs. Fe/(V + Ti) plots separate Mong Hsu ruby from other ruby fields, but did not test Mogok ruby. Thurein Taung ruby, tested here, plotted separately to Mong Hsu ruby. A Fe-Ga/Mg diagram splits ruby suites into various fields (Ga/Mg < 3), except for magmatic input into rare Mogok and Australian ruby (Ga/Mg > 6). The diverse results emphasize ruby's potential for geographic typing.

**Keywords:** ruby; Mogok; Mong Hsu; New South Wales; trace elements; LA-ICP-MS analysis; inclusions; U–Pb age-dating; genetic diversity; geographic typing

## 1. Introduction

### 1.1. Background

Corundum is an aluminum oxide mineral in which trace element substitution of Al by Fe, Ti, V and Cr act as chromophores. Rubies and sapphires are varieties in which ruby develops a red color

with sufficient entry of $Cr^{3+}$, whereas sapphire develops blue, green, yellow purple, violet, mauve and pink colors due to substitutions of the other chromophore elements in the presence of lesser Cr contents. Numerous gem corundum-bearing sites are known in SE Asia, where Mogok and Mong Hsu in Myanmar are well known examples of ruby deposits [1–3]. In contrast, East Australia is particularly noted for widespread placer sapphire deposits [2,3] and only scattered ruby-bearing deposits with rubies that are unusual in character [4]. Ruby occurs in diverse geological settings, with ages ranging from Neoarchean [5]-Proterozoic [3,6] to <5 Ma, and can form at wide ranges in temperature, pressure, and fluid activity and oxidation conditions [1–7]. The ruby deposits are primarily formed by regional metamorphism and/or from transportations from depth by volcanic processes, but the origins of the corundum can be multi-staged processes [3,4]. Eluvial and alluvial (placer) ruby deposits can form by weathering of the primary sources. Trace element studies together with O-isotope data and age dating can be used to discriminate between lithological sources [7–11]. Geochemical testing, initially leading to finger-printing [12], and further aided by O-isotope studies have become routinely used to enable geographic typing of the corundum suites [13–16]. This now achieves greater control through use of a wider range of minerals included in corundum for providing precise isotopic dating [17–19]. Quality ruby is an expensive gemstone and is mostly not favored for even micro-destructive study. Optical mineralogy combined with Raman spectroscopy remains a non-destructive technique for their testing, along with their solid/melt inclusions, for origin determination. [20,21].

In this study [22], we present new comparative trace element results and age data from Myanmar ruby fields, which typify metamorphosed and metasomatized carbonate and skarn settings at Mogok and Mong Hsu, and compare them with eastern Australia ruby fields, which typically carry ruby xenocrysts derived from basalt fields and found in placer deposits. This allows discussion of the extreme individual diversities and geochemical characteristics found within and between Myanmar ruby deposits and placer ruby sources in eastern Australia.

The regional tectonic settings of the Mogok and Mong Hsu, Myanmar and East Australian gem regions involve the western Pacific continental margins, associated with the Asian and Australian plates. The distribution of these zircon-corundum associations is shown in Figure 1, derived from references [23,24]

## 1.2. Local and Geological Settings, Myanmar

The corundum-bearing gem deposits in the Mogok area include both in situ and secondary deposits. They were described in detail [25,26] and only a relevant account is outlined here. A geological map, and associated gem workings of the Mogok area, is shown in Figure 2. The Mogok gem stone tract at the northern Mogok Metamorphic Belt (MMB) [27–32] is a source of world-class rubies, sapphires and other gemstones. Myanmar ruby genesis is associated with carbonates [33,34].

The Mogok area is characterized by high-grade metamorphic rocks; the dominant unit is banded gneiss with biotite, garnet, sillimanite and oligoclase. It is also interspersed with quartzite and bands and lenses of marble with ruby [25,26]. The metamorphic rocks are intruded by alkaline igneous rocks (mostly Oligocene-early Miocene sodic nepheline-syenite, and syenite-pegmatite and urtite suites) and early Oligocene leucogranites. Earlier Jurassic (?) to early Cretaceous mafic-ultramafic peridotites, norites minor dolerites and basalts are considered to represent layered cumulate intrusions rather than ophiolite sequences. Biotite granitoids are widely exposed at Kabaing, (16 Ma) and Thabeikkyin (130 Ma) near Mogok and some Oligocene-Miocene syenite pegmatites contain sapphires [25,26]. The Kabaing granitoids and metasedimentary rocks are commonly intruded by late-stage pegmatites and aplites [35].

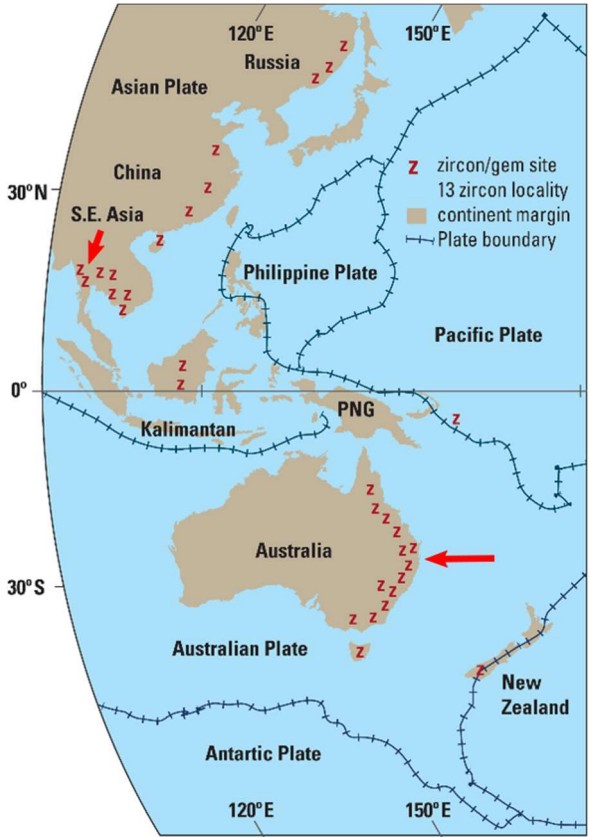

**Figure 1.** Regional setting and location of Myanmar and East Australian gem deposits, within the West Pacific continental margin zircon-corundum gem deposit zones (z), along the Asian and Australian plates [23,24]. Mogok and Mong Hsu, Myanmar, deposits (short arrow) are the western most sites [26] and New South Wales deposits (long arrow are among the eastern most group [9].

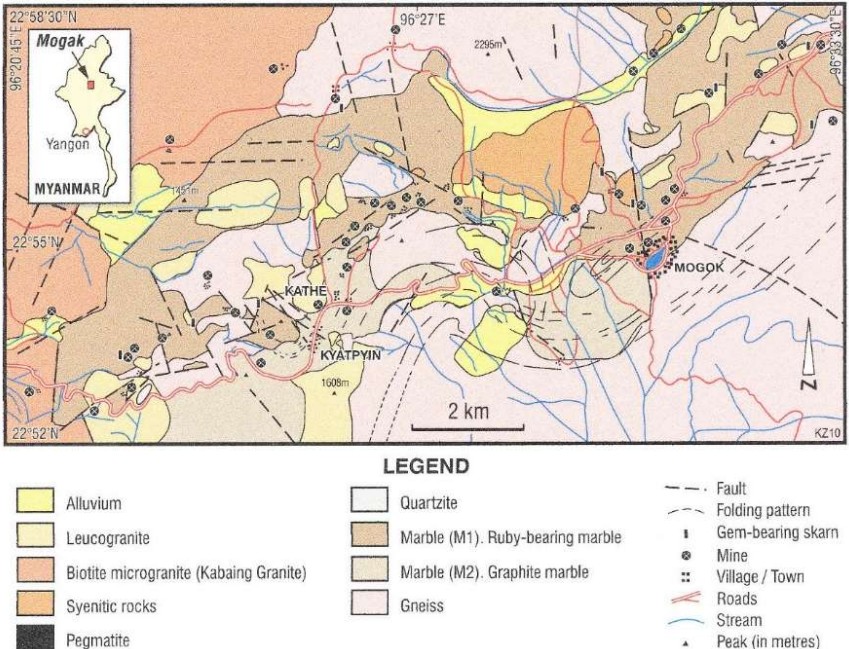

**Figure 2.** Geological map of Mogok Gem Tract, modified from reference [11], showing main lithological units and general distribution of gem deposits. The Thurein Taung workings lie ~4 km East of the 96°20′45″ E longitude map border and 2.3 km North of the 22°52′ N latitude map border.

The Thurein Taung study site is ~23 km W of Mogok Township at 96°22′ 20.7 E, 22°54′ 12.7 N. A prominent hill here has a core of steeply dipping white marble intruded by ijolite (Figure 3a,b). It is flanked by diopside-marble intruded by alkali syenite pegmatite on one side and by gneiss on the other flank. A skarn deposit in contact with leucogranite lies near the SW summit, where ruby and painite are found associated with titanite, anatase, rutile, baddelyite, axinite, elbaite, schorl, dravite and zircon [1]. The hill is covered in parts by alluvium and mine waste.

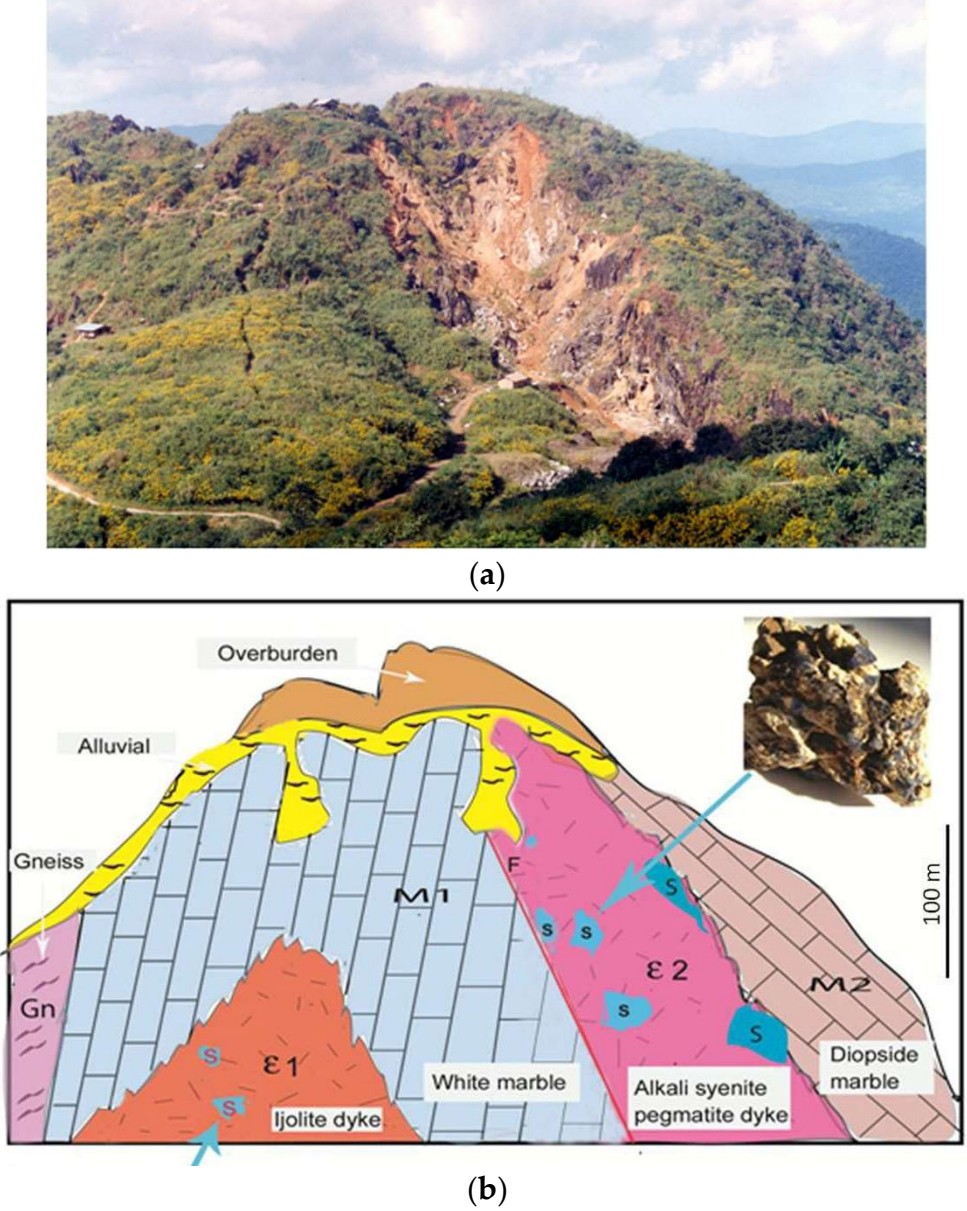

**Figure 3.** Thurein Taung ruby locality: (**a**) Hill exposing gem workings and scene for underlying geology section. Viewed from the east. Basal distance across view is ~600 m; (**b**) Sketch diagram across geological units (symbols and labels). V/H = 1, M1 = White Marble, M2 = Diopside marble, Gn = Gneiss, ε1 = ijolite dyke, ε2 = alkali syenite pegmatite dyke. Sapphire-bearing lithologies intrude the marbles (blue arrows), with sapphire deposits (s, inset). Ruby is associated with skarn near the top of the hill. Images; Kyaw Thu.

In comparison, the Mong Hsu area (Figure 4), the second-largest ruby deposit in Myanmar, is located in sedimentary and regional metamorphic rocks [17,27].

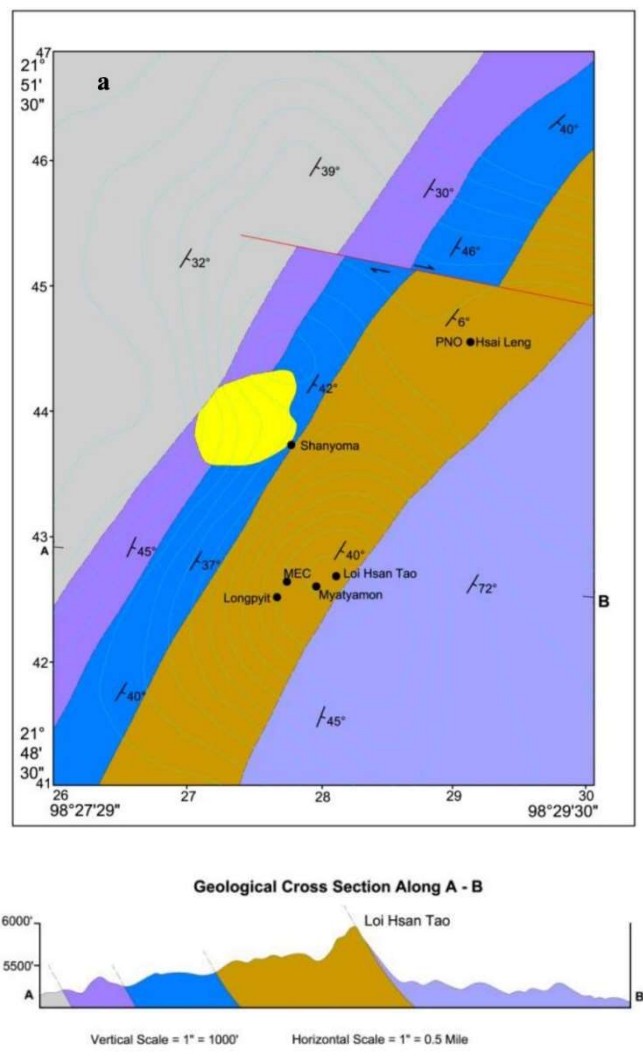

**Figure 4.** Geological map (**a**) and geological section (**A**–**B**) of Loi Hsan Tao Mountain, Mong Hsu area, Shan State, Myanmar, based on Than Than Nu in [36]. The ESE—dipping Lower Paleozoic fold belt sequence, younging from west to east includes Biotite Schist (grey), Calc-Silicates (purple), Dolomitic marble (brown), Phyllite and Quartzite (mauve). Alluvium (yellow) unconformably overlies the sequence. Structural features include strike directions (longer lines) with amount of dip (short lines normal to strike) and major fault (red line). The map north length represents 33.3 km. The ruby workings are indicated at around Lo Hasen Tao Mountain on map at cross section A–B, H/V = 2.64).

The regionally metamorphosed rocks (probably of Palaeozoic age) include biotite–almandine–staurolite schist, diopside calcsilicate rocks and biotite quartzite interbedded with biotite phyllite and ruby-bearing white marble (Figure 4). Some granitic pegmatites intruded the metamorphic rocks in the area and can be traced into the eastern and southeastern parts as far as Than Lwin River. In addition, massive Plateau Limestone (Devonian?) occupies the northern part of the area and shale, siltstone, minor limestone beds (Silurian) and bedded limestone (Ordovician) are present in the middle and southeastern parts of the Mong Hsu area.

Primary occurrences of ruby were discovered around Loi Hsan Tao Ridge (elevation 1750 m), about 3.6 km southeast of Mong Hsu town in Shan State of Myanmar [26,36,37]. In the Mong Hsu area, a series of medium grade regionally metamorphosed rocks comprised of schist, calc-silicate rock, quartzite, phyllite and ruby bearing dolomitic marble are exposed along Loi Hsan Tao ridge. The ruby occurs in a 330 m thick white fine-grained dolomitic marble within that rock sequence which dips

southeast between 30–70°. The main mode of occurrence of Mong Hsu ruby is in interconnected weak planes or fissures which run parallel to the foliation of the host marble at upper levels but become steeper and cross cut the foliation at depth. Ruby grains, lacking orientation, occur mostly as aggregates together with vein calcite. Flat crystals and half-formed crystals are frequently found. At some work sites, zonal occurrences of mineral assemblages with depth are observed, showing dolomite, tremolite and talc in the upper zone, brucite and tourmaline in the middle zone and wollastonite in a lower zone.

### 1.3. Local and Geological Settings, East Australia

In East Australian gem fields, ruby is an accessory associate of other more abundant gem xenocrysts, largely zircon and sapphire, found within scattered alluvial placer deposits [4,7,38]; Figure 5. The xenocrysts are survivors from deeper seated source rocks breached by and transported to the surface by basaltic eruptions, before erosional release into placers. In this study, rubies from three New South Wales deposits were chosen for comparison, being from diverse settings within Paleozoic fold belts in Australia [39], and representing well-characterized suites from previous studies.

At Barrington Tops, ruby occurs as fragmented and corroded composites within alluvial deposits [40], which were derived from repeated basaltic events from the long-lived Barrington Tops volcano, which had erupted for over 55 Ma [41]. The ruby-sapphirine-spinel composites suggest a metamorphic origin at 780–940 °C [42,43]. The ruby distribution within the placers overlies three separate distinctive granitoid intrusions emplaced within the underlying fold belt sequence and age-dated by zircon SHRIMP U-Pb methods at 268, 273 and 277 Ma [44]. The granitoids include mafic granodiorites formed by interactions of mantle-derived melts with crustal rocks [45]. Southwest-facing faulted slivers of relict ophiolites to the northwest include tectonized harzbugites and intrusive mafic rocks [46]. This diverse basement provides complex metamorphic, metasomatic and contact metamorphic sources for potential ruby genesis. The Macquarie River ruby grades into pink sapphire and its source region is poorly constrained, as stones were recovered as accessories in gold dredging operations in the alluvial river bed. It accompanies diamonds and represents a distinctive highly Mg-rich, high temperature genetic type [47]. The New England ruby is a rare accessory in gem field workings within the dissected Maybole shield volcano sequences [48]. It has most unusual color zoning from ruby cores into sapphire rims and consistently high Ga content and Ga/Mg ratio compared to world-wide ruby. Rubies of diverse character other than those detailed in this study are found at Yarrowitch and Tumbarumba, NSW and ENE of Melbourne and are described in references [4,9].

The rubies compared with and discussed here from the central-north New South Wales, Australian region (Figure 5) are quite different in external features to the diverse Myanmar Mogok and Mong Hsu materials and their contained inclusion suites (Figure 6a,b and Figure 7). The NSW suites lie within a NE-SW rectangular zone ~220 km in length and up to 180 km in width. They show contrasting composite mineral growth (Barrington Tops; Figure 8), high pressure mineral inclusions in pink to red gradational sapphire-ruby suites (Macquarie River; Figure 9a,b, after [4]) and unusual ruby core to multiple zoned sapphire outer zones (New England; Figure 10).

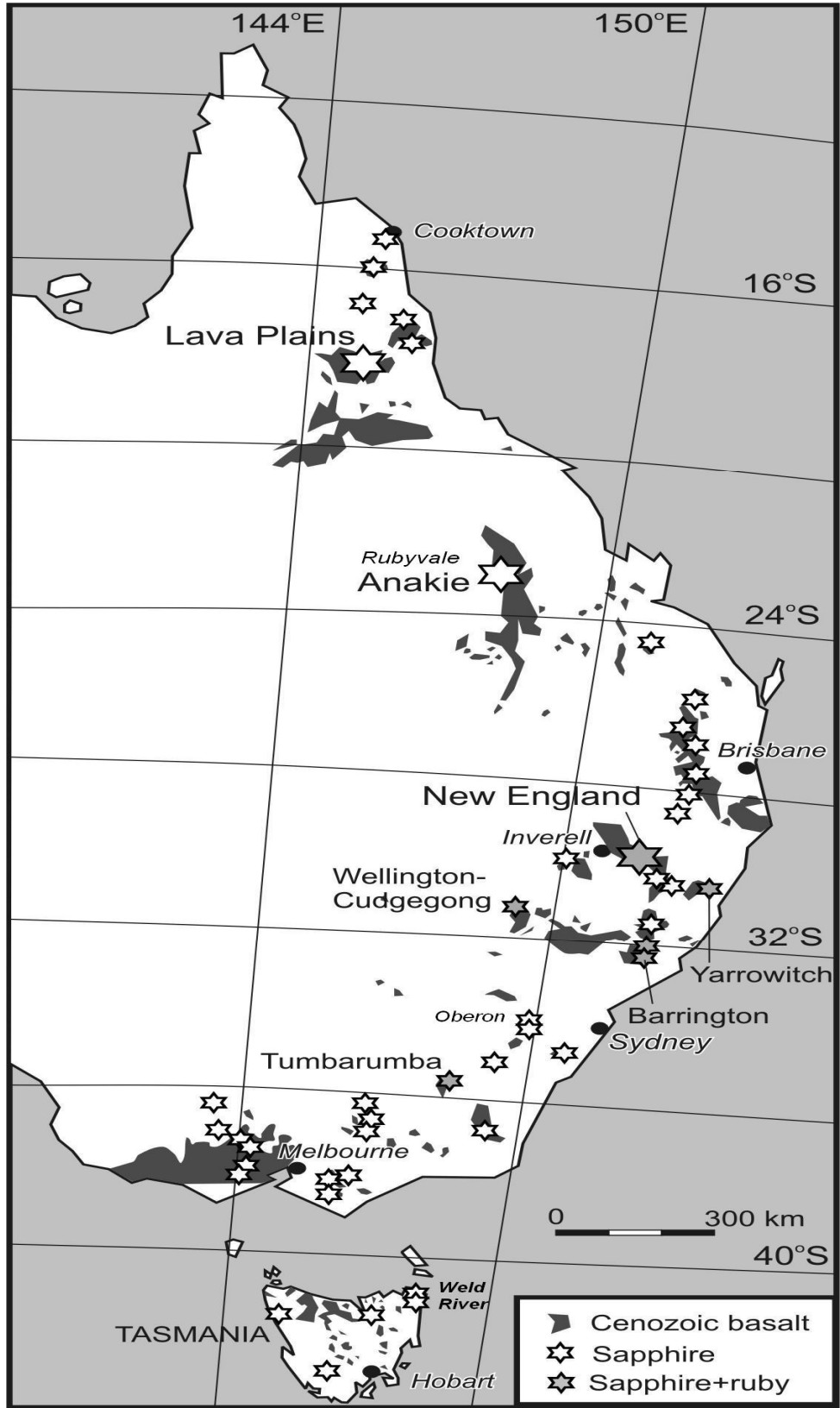

**Figure 5.** East Australian basalt-derived alluvial gem corundum deposits, showing ruby and locating Barrington Tops, Macquarie River (Wellington) and New England comparative sites. Map derived from reference [9].

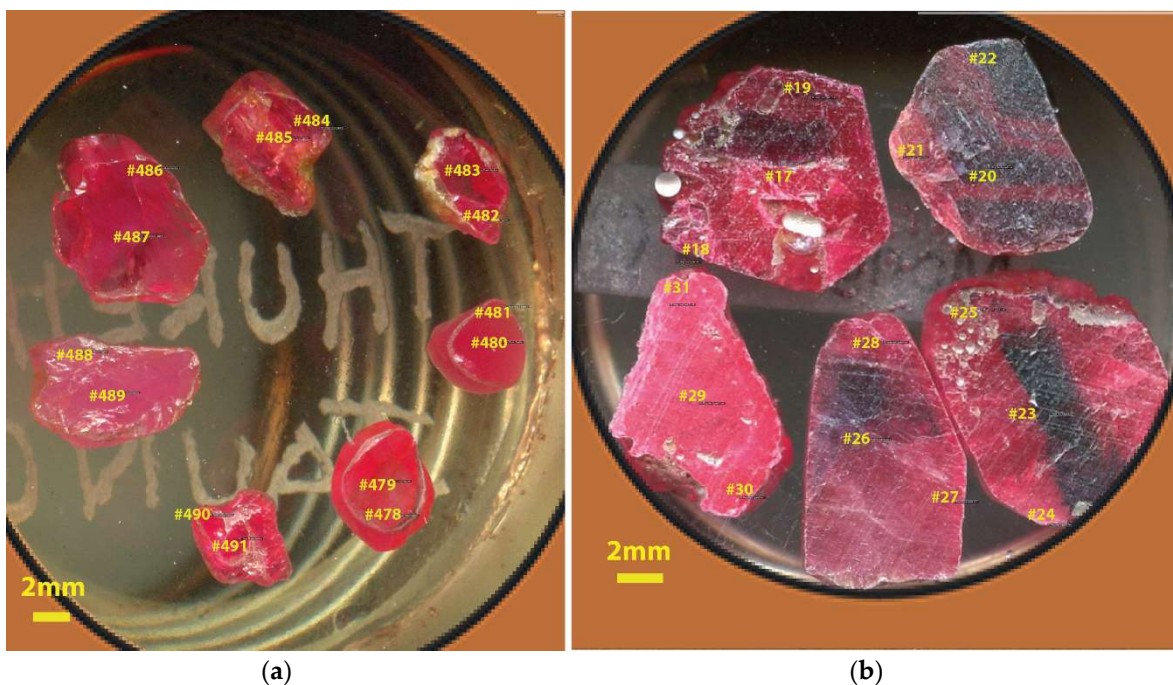

(**a**)  (**b**)

**Figure 6.** Photos showing mineralogical and textural characteristics of Mogok and Mong Hsu ruby samples. (**a**) Thurein Taung ruby crystals embedded in 25 mm diameter mount. (**b**) Mong Hsu ruby crystals embedded in 25 mm diameter mount. Yellow # nos. are analytical spots, listed in Tables A1 and A2.

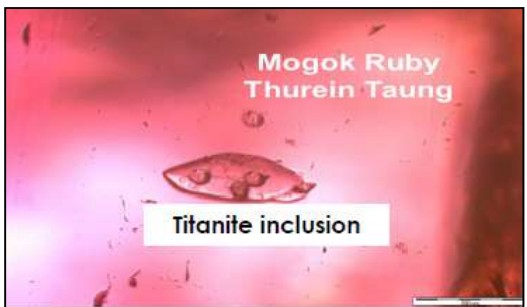

**Figure 7.** Thurein Taung ruby crystal hosting composite titanite inclusion 0.025 mm in length. The small circular structures represent ablation pits from initial analyses. Photo; Khin Zaw.

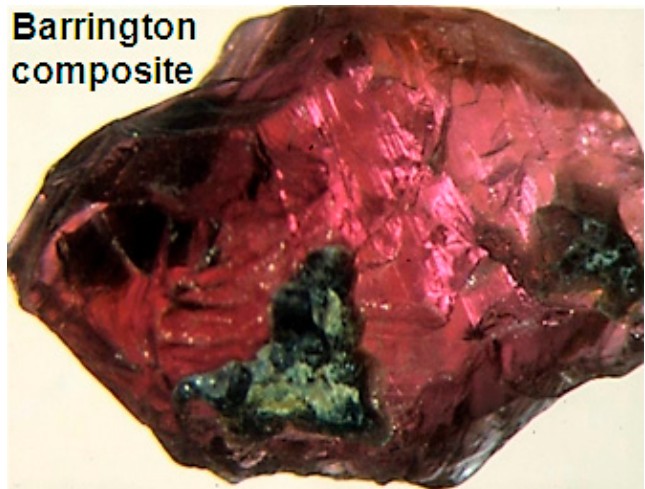

**Figure 8.** Barrington Tops ruby-sapphirine-spinel composite grain used for thermometric study [42,43]. Sample size ~5 × 7 mm. Ruby main mass (purple red); sapphirine (dark green), center bottom and center right; spinel (black), center left and outer fusion crust. Photographer G. Webb, Australian Museum.

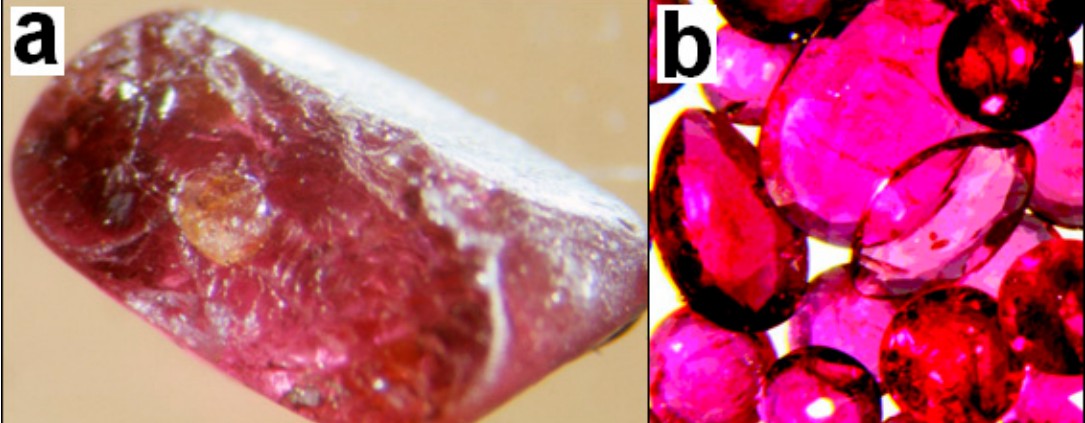

**Figure 9.** Photos of ruby features from Macquarie-Cudgegong River system alluvial deposits, NSW (**a**) ruby fragment, a few mm across with pyrope garnet inclusion (center). (**b**) Facetted stones of ruby and pink sapphire, up to 3 mm across [47]. Photographer G. Webb, Australian Museum.

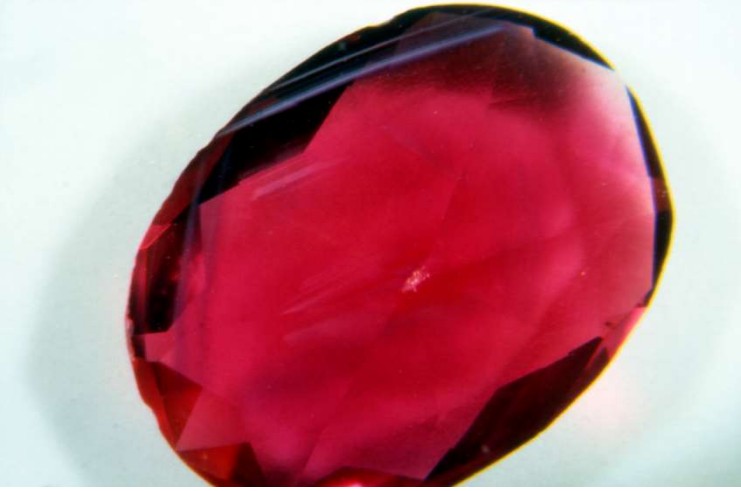

**Figure 10.** Facetted ruby (2.31c) from New England basaltic gem field [48]. Australian Museum Collection. Photo G. Webb, Australian Museum.

## 2. Materials and Methods

### *2.1. Materials*

Examples of Myanmar ruby samples analyzed for trace elements by LA-ICP-MS and an inclusion used for U-Pb age-dating are shown in Figures 6 and 7. Examples of East Australian ruby samples with previous analytical trace element data are shown in Figures 8–10.

The Australian rubies were previously analyzed from Barrington Tops and Macquarie River [9] and New England [48], and results are used for comparative plots in variation diagrams in this study.

### *2.2. Analytical Methods*

#### 2.2.1. Analytical Methods, Trace Element and Isotopic Dating Analysis

The Myanmar ruby samples and mineral inclusions used for U-Pb age dating were analyzed for trace elements contents by LA-ICP-MS techniques, supplemented by further EDS analysis (see below) at the University of Tasmania. Previously determined trace elements in New South Wales, East Australian ruby samples, were used for the comparisons had been analyzed by similar LA-ICP-MS methods at University of Tasmania (Barrington Tops, Macquarie River) and GEMOC Key Centre facilities, Macquarie University, Sydney (New England gem field). They are detailed in other publications [9,48].

#### 2.2.2. Trace Elements Analysis

Most trace element analyses were performed at CODES, University of Tasmania, using an ASI RESOLution S155 laser ablation system with Coherent Compex Pro 110 excimer laser operating at 193 nm wavelength and ~20 ns pulse width. The laser ablation system was coupled to an Agilent 7900 quadrupole inductively coupled plasma mass spectrometer (ICP-MS). Ablation was performed in pure helium flowing at 0.35 L/min and immediately mixed with argon flowing at 1.05 L/min after ablation. Depending on the session, the laser was pulsed at either 5 Hz or 10 Hz with either a 43 or a 51 μm spot size and a fluence of ~4 J·cm$^{-2}$. Each analysis consisted of 30 s of background gas, followed by 30 s of ablation time counting for 10 ms on each isotope. Primary calibration of trace elements was done with the NIST612 reference glass using GeoReM preferred values. A secondary standard correction was done using GSD-1g and BCR-2g reference glasses, again with GeoReM preferred values. Data reduction was done in an in-house macro-based Excel workbook, with aluminum as the internal standard element assuming stoichiometric proportions. Spot analyses were made at core and rim positions on each of the analyzed grains.

Backscattered electron (BSE) imaging and energy dispersive X-ray spectrometry (EDS) major elements analyses were performed at the Central Science Laboratory, University of Tasmania. This used a Hitachi SU-70 field emission scanning electron microscope (SEM, Hitachi High Technologies, Hitachinaka-shi, Japan) fitted with a Hitachi photo diode BSE detector and an Oxford AZtec XMax80 EDS system at an accelerating voltage of 15 kV and beam current of around 2 nA (Oxford Instruments Nanoanalsis, High Wycombe, UK). Elements were calibrated on a range of natural and synthetic mineral standard reference materials. Cobalt metal was used as beam measurement standard, i.e., to indirectly determine the relative change in beam current compared to the time of element calibration. The sample was coated with around 20 nm of carbon prior to analysis using a Ladd 40000 carbon evaporator (Ladd Research Industries, Williston, VT, USA).

#### 2.2.3. U-Pb Isotopic Analysis

All U-Pb analyses were conducted on the same instrumentation described above, but with the addition of $N_2$ gas for higher sensitivity. Analyses were done with a 19 μm spot size at 5 Hz and 1.9 J cm$^{-2}$ laser fluence. Each analysis consisted of 30 s of background gas, followed by 30 s of ablation time. Isotopes analyzed were $^{31}$P, $^{49}$Ti, $^{56}$Fe, $^{89}$Y $^{91}$Zr, $^{93}$Nb, $^{139}$La, $^{140}$Ce, $^{141}$Pr, $^{146}$Nd, $^{147}$Sm, $^{151}$Eu, $^{157}$Gd, $^{163}$Dy, $^{165}$Ho, $^{166}$Er, $^{169}$Tm, $^{172}$Yb, $^{175}$Lu, $^{178}$Hf, $^{181}$Ta, $^{202}$Hg, $^{204}$Pb, $^{206}$Pb, $^{207}$Pb, $^{208}$Pb, $^{232}$Th, $^{235}$U, and $^{238}$U counting for 5 ms on each isotope, except for Pb isotopes and $^{238}$U which had 25 and

15 ms counting times respectively. Each of the three minerals targeted for U-Pb analyses had a mineral specific primary standard for the U-Pb calibration: the 91500 zircon, for zircon analyses, Phalaborwa baddeleyite for baddeleyite analyses and an in-house titanite standard (19686) for titanite analyses. Calibration of the $^{207}Pb/^{206}Pb$ ratio was done using the NIST610 reference glass measured at the same conditions as the unknowns and using values from [49]. Calibration of select trace element data measured along with Pb/U ratios was done using the NIST610 reference glass using GeoReM preferred values and with Zr as the internal standard element for baddeleyite and zircon and Ti as the internal standard element for titanite; all assuming stoichiometric proportions in each mineral respectively. Data reduction for U-Pb and trace element concentrations is done in an in-house macro-based Excel workbook with details about the methodology described in reference [50].

Accuracy of the U-Pb ages was checked using a variety of reference minerals of known (ID-TIMS) age treated as unknowns. These include the Temora [51] and Plesovice [52] zircons, FC-1 baddeleyite [53], and the FCT-3 [54] and 100606 [55] titanites. All of these secondary reference materials were accurate within their uncertainties. Concordia plots and intercept ages are done using Isoplot 4 [56]. Where a common Pb estimate is lacking, the Stacey and Kramer [57] model Pb at the age of the mineral was used for the anchor on the Concordia intercept ages. Uncertainties are calculated using the method of Horstwood et al. [58], where systematic uncertainties are added after the Concordia intercept ages are calculated. Details of the geochronologic data are supplied in Supplementary Materials.

## 3. Results

### 3.1. Trace Element Variations

The main trace element values measured in Thurein Taung and Mong Hsu ruby samples are given in Table 1. This breaks the data up into rim and core measurements of the analyzed crystals for easier comparison, while individual results from each analytical spot are detailed in Tables A1 and A2. Some rubies appeared to indicate noticeable Si and Ca values above the relatively high detection limits for those elements, particularly in Thurein Taung ruby. Such levels that enter the corundum crystal structure invoke potential presence of nano inclusions of silica [59] or even particulate calcsilicate material [13]. Because of potential large variability in Si and Ca within corundum analyses, due to LA-ICP-MS analytical interreference affects [60], these element values were not reported in Tables 1, A1 and A2. Other trace elements are mostly at detection level or negligible (1 or <1 ppm) in concentration. A few Thurein Taung rubies contain 1–2 ppm B, while Mong Hsu rubies contain 1–3 ppm Cu and 1–9 ppm Zn.

**Table 1.** Trace element comparative ranges and averages (ppm), Myanmar ruby samples.

| Sample | Mg | Ti | V | Cr | Fe | Ga |
|---|---|---|---|---|---|---|
| **Thurein Taung** | | | | | | |
| Rims, range | 27–204 | 38–1241 | 110–401 | 432–4599 | 38–483 | 20–170 |
| n = 9, average | 85 | 254 | 243 | 2468 | 143 | 79 |
| Cores, range | 35–189 | 53–283 | 102–421 | 618–4406 | 24–461 | 20–152 |
| n = 9, average | 78 | 139 | 225 | 2494 | 127 | 75 |
| **Mong Hsu 1** | | | | | | |
| Rims, range | 27–75 | 336–3201 | 73–628 | 429–3545 | 10–32 | 48–78 |
| n = 7, average | 48 | 1410 | 391 | 2026 | 14 | 69 |
| Cores, range | 42–95 | 1168–2550 | 75–649 | 919–4625 | <8–17 | 48–80 |
| n = 4, average | 70 | 1815 | 324 | 3199 | <9 | 66 |
| **Mong Hsu 2** | | | | | | |
| Rims, range | 20–147 | 42–1492 | 183–1012 | 959–16,388 | 11–51 | 68–105 |
| n = 14, average | 68 | 778 | 342 | 5600 | 29 | 83 |
| Cores, range | 41–226 | 100–2633 | 228–658 | 1194–27,386 | 15–54 | 81–103 |
| n = 8, average | 115 | 1210 | 417 | 7015 | 26 | 89 |

Data based on detailed analyses listed in Appendix A Tables A1 and A2. < = value bdl.

The main chromophore Cr is distinctly higher in Mong Hsu set 2 rubies (max. ~2.7 wt %, av. 0.65 wt %), than in Mong Hsu set 1 (max. 0.45 wt %, av. 0.23 wt %) and Turein Taung (max. 0.46 wt %, av. 0.25 wt %) rubies. One set 2 zoned crystal is significantly more enriched in Cr (max. 2.7 wt %, av. 1.9 wt %) than for 7 others in the set (max. 1 wt %, av. 0.44 wt %). The most variable chromophore is Ti, which is lowest in Thurein Taung rubies (max. 1250 ppm, av. 265 ppm) compared with Mong Hsu set 1 (max. 3200 ppm, av. 1362 ppm) and Mong Hsu set 2 (max. 2633 ppm, av. 935 ppm) rubies.

In Turein Taung rubies, average rim ppm values in Cr are less than average core values (− values), but are greater in Mg, Ti, V, Fe, Ga rim values (+ values). The rubies, however, are relatively homogenous with rim to core zones showing limited ppm variations (Mg +7; Ti +115, V +18; Cr −26; Fe +16; Ga +4). In contrast, wider average zonal spreads appear in ruby sets Mong Hsu 1 (Mg −22; Ti −405; V +67; Cr −1173; Fe + >9, Ga +3) and Mong Hsu 2 (Mg −47; Ti −432; V −75, Cr −1415; Fe +3; Ga − 6).

Thurein Taung V values at 102–421 ppm lie well below more extreme V values at 900–5500 ppm in nearby parts of the Mogok ruby field [11,13]. Thurein Taung Cr/V ratios, however, range up to 37 and suggest links with the Mogok V-rich ruby province. In comparison with Thurein Taung, the Mong Hsu ruby sets include higher V values up to ~1012 ppm and range into even higher Cr/V ratios (Mong Hsu 1, 1.2–37.1, av. 11.3; Mong Hsu 2, 4.0–59.5, av. 18.0).

*3.2. U-Pb Ages*

All zircon and titanite analyses were done in-situ, as small grains present as inclusions within corundum megacryst hosts. Baddeleyite was analyzed as large euhedral megacrysts. All age results and all secondary reference material analyses are presented in Supplementary Materials. Zircon and ilmenite inclusions were targeted for U-Pb analysis in the Mong Hsu rubies. These inclusions were generally >30 μm in size and euhedral in shape. Ilmenite analyses contained modest U contents (<70 ppm), however, they contained significant common Pb and so were useful as a common Pb anchor point for the zircon analyses. The zircons have significantly high U, up to 3000 ppm, and high U/Th ratios (up to 40). Analyses show significant variation in common Pb corrected age (207Pb correction), often with variable Pb/U ages within single analyses. A Concordia intercept age of 23.9 Ma (± 1.0/1.03 Ma, including systematic uncertainties respectively), calculated on the four youngest zircons formed a coherent population (MSWD 1.10, probability of fit 0.35) with ilmenite analyses included for use as a common Pb anchor (Figure 11).

Zircon and titanite inclusions were also targeted for U-Pb analysis in the Thurein Taung ruby. The zircons were generally <30 μm in size, while the titanite was a single grain ~250 by ~150 μm in size and euhedral in shape. Multiple analyses were done on this single grain to constrain the age as no other titanite grains were exposed at the surface. A Concordia intercept age of 32.34 Ma (± 0.97/1.03 Ma) was calculated on the 7 titanite analyses on this single grain with the age anchored on a $^{207}Pb/^{206}Pb$ ratio of 0.837 (Figure 12). Three zircons included in the same ruby megacryst as the titanite give a range of older ages from ~50 to ~100 Ma, despite being a few millimeters away from the titanite.

Two baddeleyite megacrysts from the Thurein Tuang, ~5 × 10 mm in size, were analyzed targeting both rims and cores. A wide range of ages were measured in the two megacrysts from ~40 Ma to ~110 Ma. Eight of the analyses formed a coherent age population at 103.3 Ma (± 2.2/2.45 Ma). There was no obvious correlation with the measured age and location (core vs. rim) of the baddeleyite, with some rims containing the ~103 Ma age population and some cores containing significantly younger ages. Likewise, there is no correlation with the trace element chemistry of the baddeleyite and the measured ages.

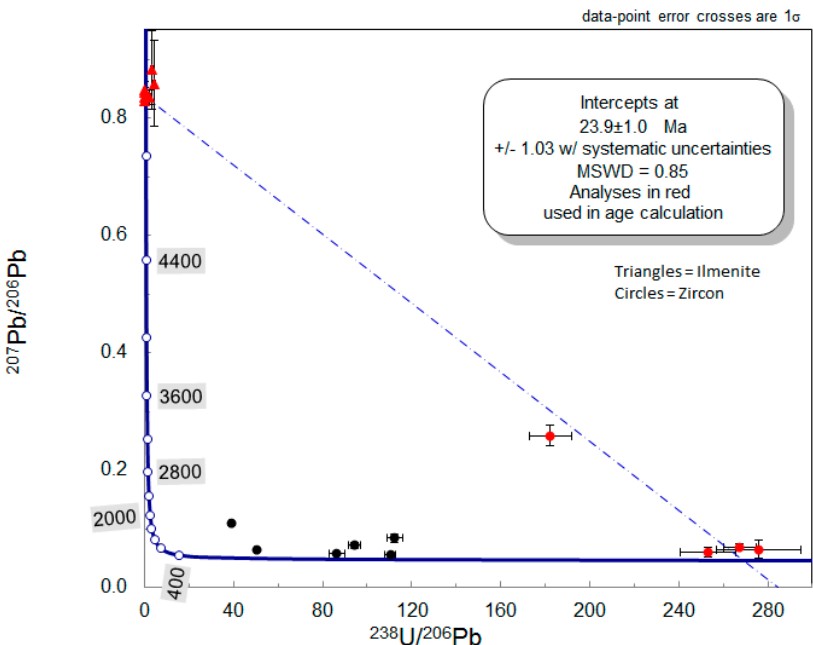

**Figure 11.** Concordia diagram showing zircon plots (red spots) used to constrain a concordia age of the four youngest zircons within the Mong Hsu host ruby. Age measurements on two baddeleyite megacrysts from the site (grey spots) are shown, but relationships to Mong Hsu ruby are uncertain.

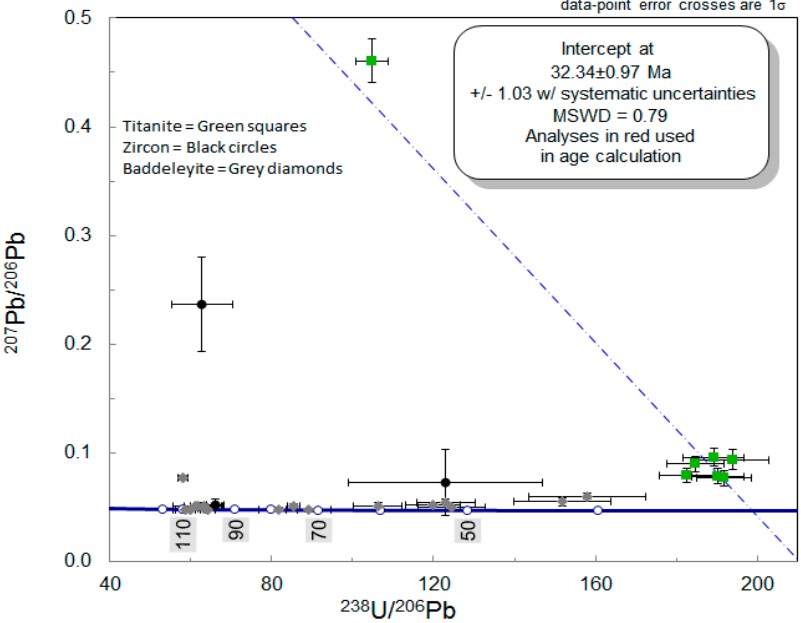

**Figure 12.** Concordia diagram showing titanite plots (green spots) and error bars constraining the intercept age of the titanite composite inclusion in Thurein Taung ruby. Comparative age plots are shown for older zircon included ruby (grey) and baddeleyite megacrysts (black) from Thurein Taung.

Trace elements in the young 23–27 Ma zircon in Mong Hsu ruby were analyzed by LA-ICP-MS during age-dating (Supplementary Materials). Three zircons showed consistent results, based on an assumed Zr content of 493,000 ppm, with Hf 17,334–18,659, U 2012–3407, Th 155–241, U/Th 13.6–14.1, Y 99–161 and very low LREE-MREE and enriched HREE. One zircon contains a Ti-rich inclusion, probably rutile that skewed results, giving excessive Ti and reducing other values. The ilmenite used as an anchor for common Pb ranged in Ti/Fe ratios between 0.47–3.73 and contained minor amounts of P, Zr, Nb and LREE.

*3.3. Mineral Inclusion Analyses*

Mineral phases distributed in the dated composite titanite inclusion and dated zircon xenocrysts within host Thurein Taung ruby were targeted for EDS analysis using BSE images (Figures 13 and 14; Table 2). The composite inclusion is an elongated subhedral, titanite crystal ~270 μm long, intergrown with two euhedral nepheline crystals, 30 and 65 μm across, on its margins that constitute ~4% of the inclusion. Mineral formulas given in Table 2 were based on O calculated by cation stoichiometry, with titanite standardized on Si.

Detailed LA-ICP-MS trace elements including REE were acquired for the Thurein Taung composite titanite and zircon inclusions (Supplementary Materials). Titanite analyses (n = 7), based on an assumed Ti content of 181,607 ppm, contain accessory ppm of P (300–837), Fe (134–171), Y (555–583), Zr (1100–1997), Nb (828–890), Ta (53–59), Th (651– 733), U (248–3206), and have av. Th/U (2.6). Analyses are moderately elevated in LREE-HREE (La–Lu 1726–1906). Zircon analyses, using assumed Zr of 493,000 ppm, are enriched in Hf (16,890–19,504), show minor elevations in Ti, (27–61), Fe (<14–165), Y (57–66), Th (22–54), with av. Th/U (0.35). They are noticeably low in LREE-MREE but show mild enrichment in HREE. Baddeleyite analyses, using assumed Zr of 740,000 ppm, are enriched in Hf 16890–1950), have elevated Ti (1537–2813), Nb (308–1147), Ta (262–525), minor P (117), Pb (19–118) and negligible REE and Th.

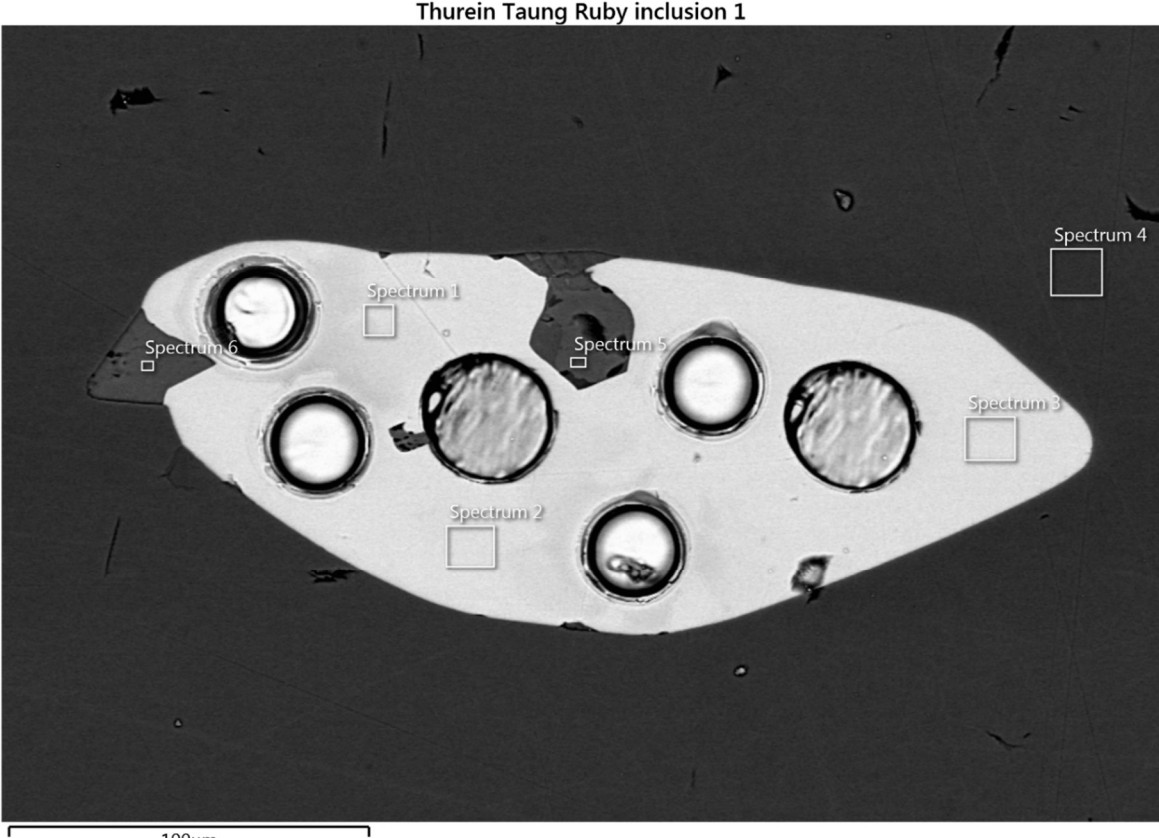

**Figure 13.** BSE image of titanite-nepheline inclusion (light grey) in Thurein Taung ruby (dark grey background). Darker euhedral indents in the inclusion (top and left) are nepheline. Circular pits mark age-dating sites, while EDS spectrum sites (rectangular boxes) represent titanite analyses, listed in Table 2).

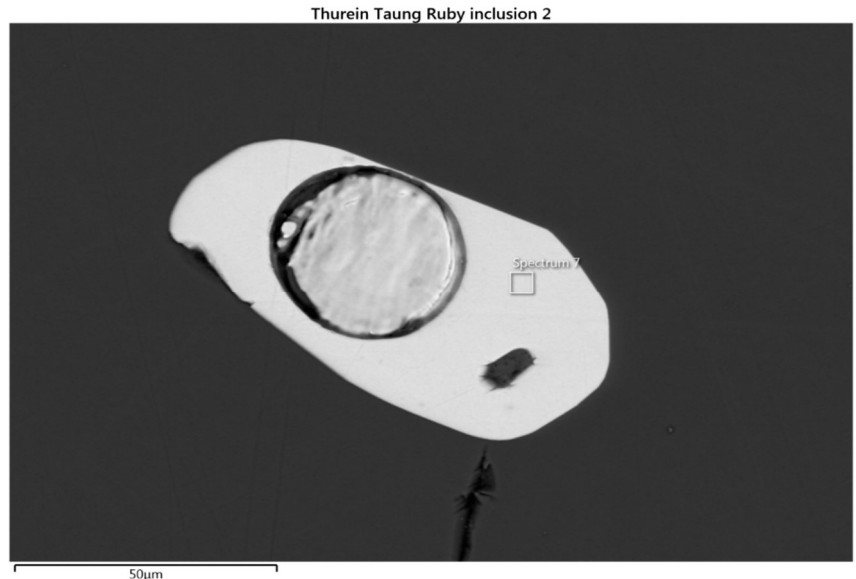

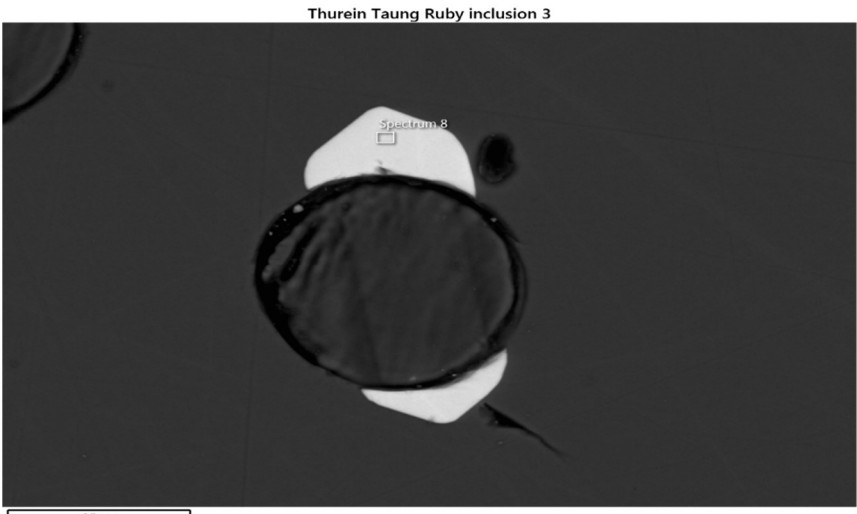

**Figure 14.** Zircon crystals in Thurein Taung ruby (background), showing circular age-dating pits and EDS spectrum sites (rectangular boxes) represented in analyses listed in Table 2.

**Table 2.** EDS element analyses (wt %), inclusion mineral phases, Thurein Taung ruby.

| **Titanite (Composite Inclusion)** [1] $Ca_{4.00}$ $(Ti_{3.35}, Al_{0.68})_{4.03}$ $Si_4$ $(O_{19.72}, F_{0.55})_{20.27}$. | | | | | | |
|---|---|---|---|---|---|---|
| Element (σ) | O (0.4) | Si (0.1) | Al (0.1) | Ca (0.1) | Ti (0.1) | F (0.1) |
| Av., n = 3 | 41.7 | 14.2 | 2.3 | 20.3 | 20.3 | 1.3 |
| **Nepheline (composite inclusion)** [2] $(Na_{0.69}, K_{0.07}, Ca_{0.08})_{0.91}$ $(Si_{1.1} Al_{0.9})_{2.0}$ $O_4$. | | | | | | |
| Element (σ) | O (0.2) | Si (0.1) | Al (0.1) | Ca (< 0.1) | Na (0.1) | K (< 0.1) |
| Av., n = 3 | 45.3 | 20.9 | 17.9 | 2.2 | 11.2 | 1.8 |
| **Zircon 1 (xenocryst)** [3] $(Zr_{1.06}, Hf_{0.01})_{1.07}$ $Si_{01.05}$ $O_4$. | | | | | | |
| Element (σ) | O (0.2) | Si (0.1) | Al (0.0) | Zr (0.3) | Hf (0.2) | |
| Av., n = 1 | 32.4 | 14.9 | bdl | 48.8 | 1.2 | |
| **Zircon 2 (xenocryst)** [4] $(Zr_{1.00}, Hf_{0.01})_{1.01}$ $Si_{0.99}$ $O_4$. | | | | | | |
| Element (σ) | O (0.1) | Si (0.1) | Al (0.0) | Zr (0.3) | Hf (0.2) | |
| Av., n = 1 | 31.4 | 14.9 | bdl | 48.1 | 1.2 | |

[1,2,3,4] Calculated mineral formula of analyzed mineral. bdl below detection limit.

## 4. Discussion

The gem mineralization associated with the Myanmar ruby deposits is inherited from the stratigraphy and dynamic tectonic events that built the geological framework within the gem tracts [59,61,62]. The initial discussion considers the age-dating results on inclusions within Myanmar rubies from the two studied deposits and examines their relationships to previous dating in ruby deposits in SE Asia. This is followed up by synthesizing the trace element results from the Myanmar sites and Australian ruby suite data within a range of genetic classification diagrams. With characterization, the suites will be examined for their geochemical traits that enable distinctions between these suites. Finally, comparative elemental 'fingerprinting' is considered in a broader context of geographic typing of rubies within a global perspective.

### 4.1. Mogok and Mong Hsu Ruby Ages

The U-Pb age of the Thurein Taung composite titanite inclusion at 32.4 ± 1 Ma, matches the age of small leucocratic granite bodies that intrude the area [25]. The near-euhedral crystal form suggests a likely syngenetic origin and a host ruby age linked to alkaline melts and skarn formation associated with that leucocratic granite event. The presence of subordinate nepheline in the titanite composite, however, needs consideration in respect to a potential under-saturated alkaline source. Nepheline-bearing rocks are known to intrude and be faulted against the local marbles in this vicinity [25]. These urtite-ijolite members and syenite rocks, however, seem younger than the composite inclusion age. This is based on a 25 Ma age for a foliated syenite and observations that the syenites themselves are intruded by the urtites and by pegmatites associated with the Kabaing granite, dated at 16.8 ± 0.5 Ma [25,61]. This, in conjunction with a zircon U-Pb age of 16.1 Ma in a painite overgrowth on ruby from Wetloo [25], suggests multiple periods of ruby growth may have taken place in the Mogok skarn environment [63]. This aspect is further supported by Ar-Ar age dating of phlogopite mica in syngenetic growth with ruby in the Mogok field at 17.1 ± 0.2 Ma–18.7 ± 0.2 Ma [33]. U-Pb dating of zircon inclusions is rare in constraining Mogok ruby ages, but has been used on syngenetic zircon to date quality Mogok sapphires [64]. Mogok sapphires have links with under-saturated alkaline intrusions [25]. Age dates obtained so far for host sapphires [64] gave 26.7 ± 4.2 and 27.5 ± 2.80), close to, but just outside error for the Thurein Taung composite inclusion age (this study).

Mong Hsu zoned ruby-sapphire crystals are found in host veins composed of calcite, a Mg-rich chlorite group member, Cr-bearing muscovite and opaque oxides, which traverse recrystallized dolomitic marble [37], or occur in disseminated grains in a dominant dolomite matrix with minor calcite and intergranular phlogopite [65]. Solid inclusions within the rubies themselves identified by laser Raman spectroscopy included Cr-bearing muscovite, paragonite-margarite solid solution, Mg-rich chlorite, rutile, quartz, dolomite and diaspore [66]. In addition, the present study identified zircon and ilmenite which enabled U-Pb age determination of the host ruby age. The Mong Hsu ruby age formation, based on the young zircon inclusions, at 23.9 ± 1.0 Ma is clearly younger than the Thurein Taung titanite inclusion age by some 8–9 Ma. The later age for Mong Hsu ruby generation fits in with a declining phase of regional metamorphism [36,61], within fluid-rich conditions which included $H_2O$, $CO_2$, F and other components [37]. An isochore estimation from fluid inclusion compositions within the ruby gave a P-T range of 0.20–0.25 GPa and 500–550 °C for the fluid activity, which may represent hydrothermal input from deeper magmatism [36]. This low P-T range for ruby growth was questioned from zircon-in-rutile thermometry on an inclusion in a ruby. The estimated T between 615–690 °C suggested upper amphibolite facies T in the 0.4–0.6 GPa range [66]. The present U-Pb zircon dating now yields a firmer early Miocene 24 Ma age for this metamorphic/magmatic event.

### 4.2. Myanmar and East Australian Ruby Trace Element Comparisons

Mong Hsu ruby crystal growth and trace element analysis was used to establish a geochemical identity [66]. The study recorded dark cores, which were elevated in Cr, V, Mg, Ta, W and Th

values. This trend also was noted here, in a core significantly enriched (ppm) in Cr 10,036, Ti 1272, V 909, and Mg 1272, although Ta, W and Th remained low. The earlier study suggested a distinctive geochemical field for Mong Hsu ruby, compared with other ruby fields, when Cr/Ga was plotted against Fe/(V + Ti). Strangely, however, the compared fields did not include Mogok ruby. To further test the claim, ruby analyses from Mong Hsu, Thurein Taung, Mogok, from this study and data from New England, East Australia [48] are compared with the proposed Mong Hsu field in Figure 15.

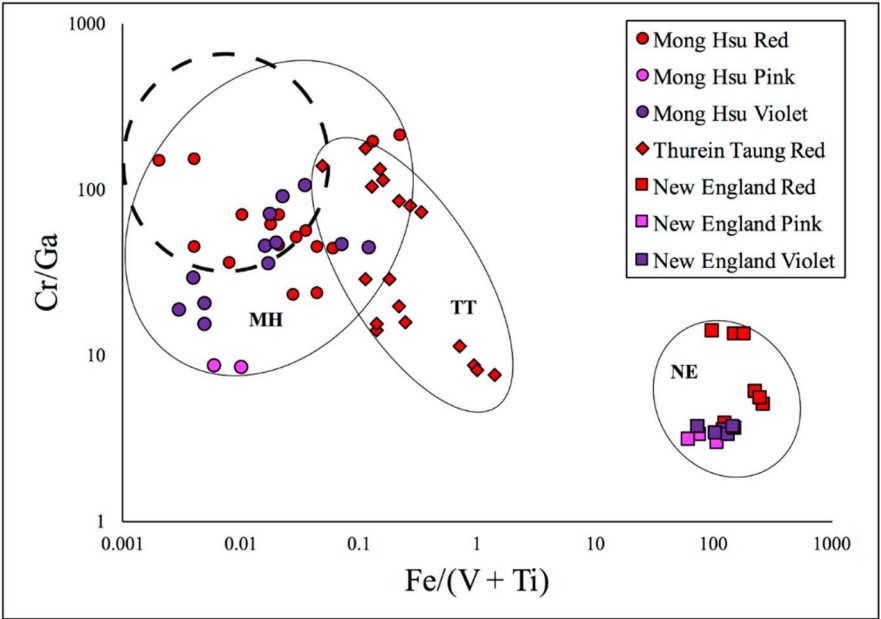

**Figure 15.** Ruby plots of Mong Hsu (MH), Thurein Taung (TT), Myanmar, and New England, East Australia (NE) are shown within oval enclosing field boundaries in a Cr/Ga vs. Fe/(V + Ti) diagram. The plots are color- and locality- coded to indicate multi-zoned crystals (top right legend). The Mong Hsu ruby field designated by Mittermayr et al. [66] is indicated by a dashed circle.

The Mong Hsu ruby plots (red zones) from the present study largely occupy the defined limits for the proposed Mong Hsu field. The violet-blue zones, however, mostly lie outside the field with lower Cr/Ga values, while pink variants show even lower Cr/Ga. Thurein Taung plots suggests some separation between two subset components. The diagram shows good separation between Mong Hsu and Thurein Taung ruby fields and extreme separation from the New England field. These trace element ratios thus appear effective in establishing a Mong Hsu ruby identity.

Further relationships between Myanmar and East Australian ruby suites are explored based on element oxide indices that classify corundum into genetic fields related to host rock type associations [3]. Comparative results are shown in Figures 16–18. In Mong Hsu rubies red zones mostly plot in the ruby-in marble field, dark violet-blue zones mostly fall in the mafic-ultramafics field and pink to white zones lie near the boundary region. These plots correspond to the R, V and I zones described by Peretti et al. [37], considered to mark repeated influx of fluids rich in F. The F interaction decreased the Ti chromophore activity through a complexing process and effectively raised Cr chromophore ratios to color red zones. The new Mong Hsu plots support multiple interplay between separate source components, involving marble host lithologies and invading fluids from deeper seated magmatic/metamorphic sources to generate multiple color zoned corundum. The Thurein Taung plots, in contrast, nearly all closely group in the ruby-in marble field, which suggests a simpler origin.

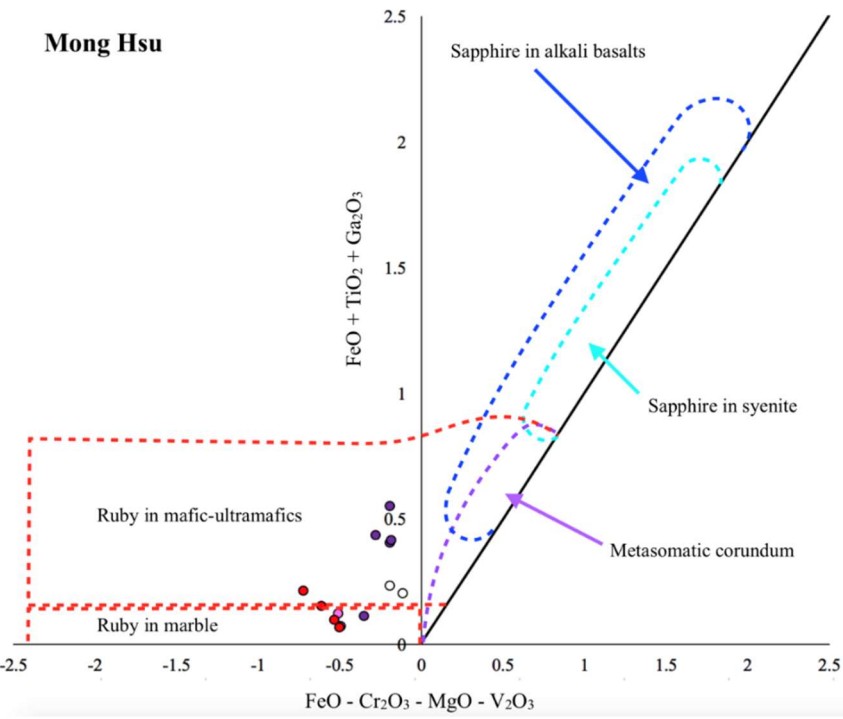

**Figure 16.** Discrimination metal oxide plots, with petrological fields after [3], for Mong Hsu ruby analyses. The plots are color-coded in relation to red, violet-blue-black and white zoning.

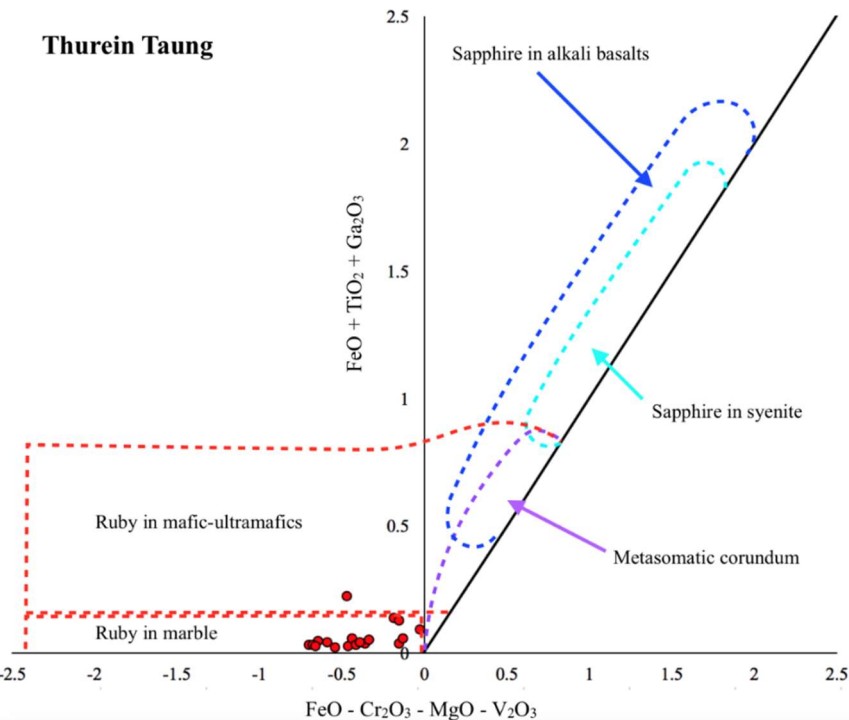

**Figure 17.** Discrimination metal oxide plots, with petrological fields after [3], for Thurein Taung ruby analyses. The plots are color-coded.

East Australian plots (Figure 18) present varied lithological affinities. Barrington Tops ruby has metasomatic corundum affinities. Macquarie River ruby in contrast forms a tight linear array near the base of the Ruby-in-mafic-ultramafics field, while the multiple color zone components of New England ruby spread along the Metasomatic corundum field to slightly overlap the Ruby-in-mafic-ultramafics

field. Myanmar ruby fields have strong connections with carbonate sources, unlike East Australian ruby sources, although both groups include some mafic-ultramafic source inputs.

　　A trace element variation diagram, introduced as a tool to separate blue-colored sapphires of metamorphic, transitional and magmatic origin, by Peucat et al. [67], plotted Fe (ppm) content against Ga/Mg ratio. This tool received wider use being extended to sapphire of other colors and even ruby, as it is normally a metamorphic/metasomatic corundum and seemed to conform to the classification boundaries. [9,68,69]. Recent studies show that individual corundum crystals can show significant ranges in their Ga/Mg ratio [70], and that rare rubies from Myanmar and East Australia show high Ga/Mg ratios that lie in the magmatic field [13,48]. Furthermore, some sapphire/ruby suites with low Ga/Mg, previously thought metamorphic can contain melt inclusions suggesting a magmatic origin [71]. This makes the Ga/Mg ratios considered in isolation unreliable genetic indicators. Plots of Fe (or Fe/Mg) against Ga/Mg, nevertheless, remain useful in aiding distinctions between fields from different localities (Figure 19).

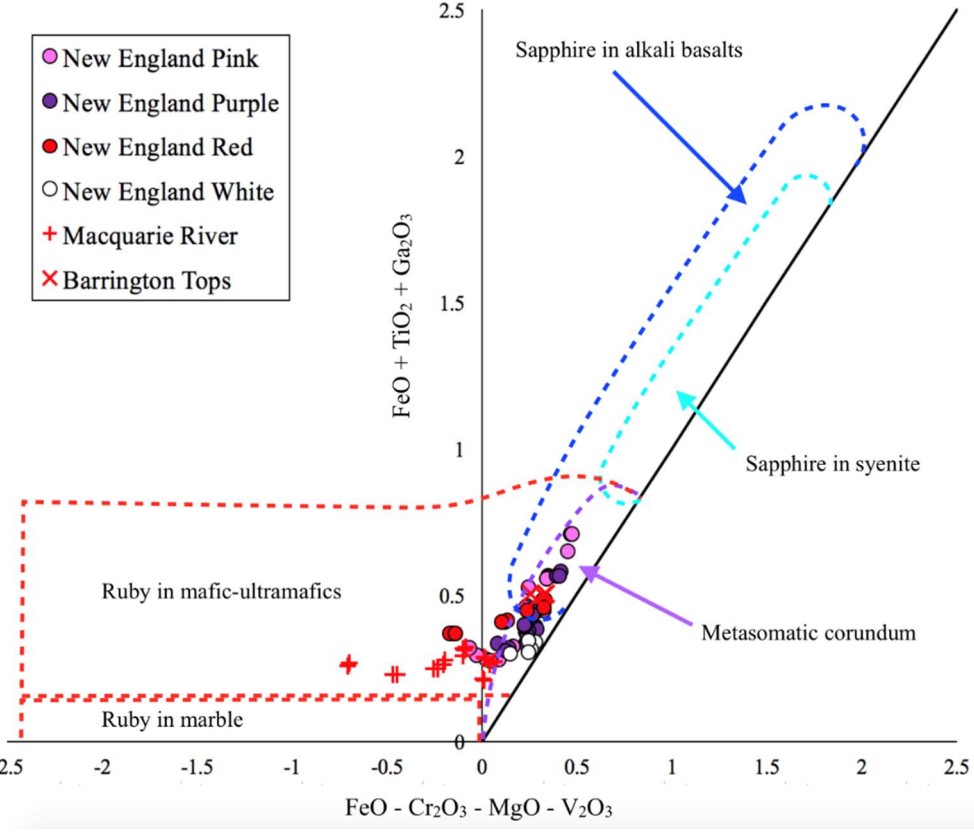

**Figure 18.** Discrimination diagram of metal oxide plots, within petrological fields after [3], for East Australian plots from Barrington Tops, Macquarie River and New England rubies. Plots are color coded to show the range of color and zoning. Symbols for different localities are shown in the legend.

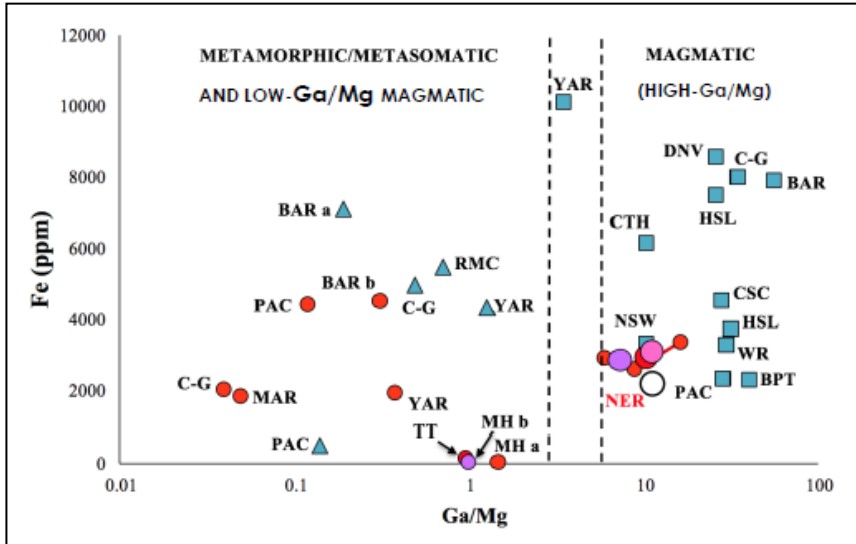

**Figure 19.** Fe-Ga/Mg diagram after [55], showing averaged plots from selected examples of ruby and sapphire suites from East Asian, East Australian and other localities, from data after [9,11,59,72]. Short dash lines mark a transitional zone (Ga/Mg 3–6) for Metamorphic/Magmatic fields [66]. Symbols include: Red-filled circles, ruby; Red-violet-white filled circles, zoned ruby-sapphire suites; Blue-filled triangles, low-Ga meta-sapphire suites; Blue-filled squares, higher-Ga sapphire suites. East Asian locality identity letters include: BT, Bo Phloi, Thailand, sapphire; CTH, Chanthanburi, Thailand sapphire; CSC, Changle, Shadong, China, sapphire; DNV, Dak Nong, Vietnam sapphire; MH, Mong Hsu, Myanmar; zoned ruby-sapphire suite; PAC, Pailin (west), Cambodia, ruby and sapphire suites; TT, Thurein Taung, Myanmar, ruby suite. East Australian locality identity letters are: BAR, Barrington Tops, magmatic sapphire, (a) meta-sapphire (b) ruby; C-G, Cudgegong-Gulgong, ruby, sapphire; MAR, Macquarie River, ruby, meta-sapphire, high-Ga sapphire; NSW, New South Wales, high-Ga sapphire; NER, New England, zoned ruby-sapphire; YAR, Yarrowitch, ruby, meta-sapphire, transitional sapphire, WR, Weld River, Tasmania, high Ga-sapphire. Other: RMC, River Mayo, Colombia, South America sapphire.

The studied Thurein Taung and Mong Hsu, Myanmar suites (Figure 19) show very low Fe and cluster at Ga/Mg ~1–2. Note that dark violet zones of Mong Hsu ruby-sapphire (MHb) separate into a slightly lower Ga/Mg group than for ruby zones. These carbonate- interacting Myanmar ruby suites are quite distinct in values from West Pailin, Cambodian ruby suite (PAC). East Australian ruby suites, in contrast, have a more extended separation and diversity than the plotted SE Asian ruby suites, extending in Fe values from ~1800–4200 ppm and Ga/Mg range from ~0.054 to 25.

Thurein Taung ruby is just one locality among many ruby deposits in the Mogok gem tract for which LA-ICP-MS analyses are available for comparison [11,13,59]. One Mogok site, Le-U, has ruby notably high in Si (1060–4290 ppm), trace B (10–35 ppm) and Sn (2–18 ppm) and $\delta^{18}O$ (20.4 mil %), and was attributed to a skarn origin [13]. This ruby is also exceptional in its high Ga contents (~370–720 ppm) and Ga/Mg ratios (46–521), exceeding those of the New England, Australian high-Ga ruby (compare Figures 19 and 20). In a further variation diagram, Le-U ruby with V + Cr (2270–31,890 ppm) plotted against Fe + Ti (4–12 ppm) falls in a distinct field outside other Mogok site plots (Figure 21). This is quite dissimilar to the Thurein Taung data (V + Cr 542–4744 ppm; Fe + Ti 82–2295, which extends across the general Mogok High-V metamorphic ruby field.

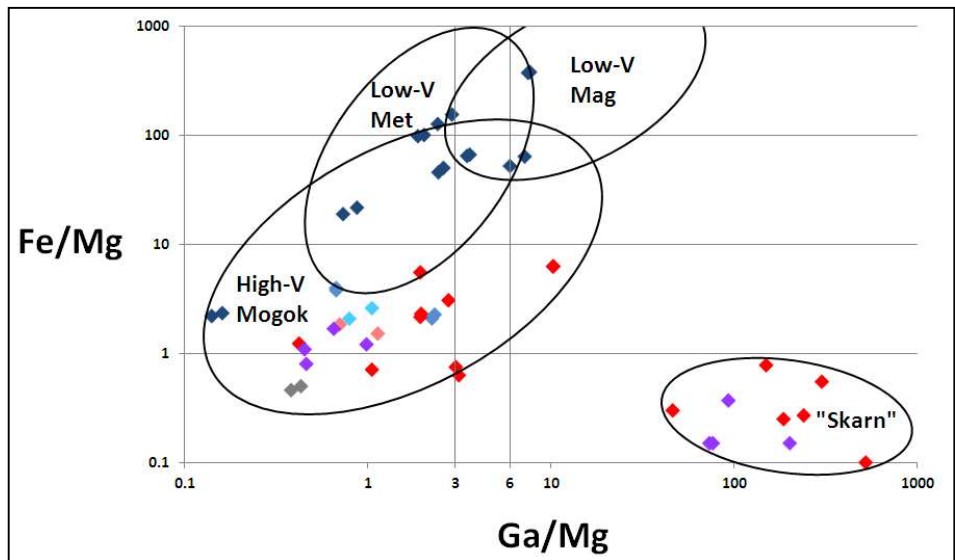

**Figure 20.** Fe/Mg vs. Ga/Mg plots Mogok corundum, as color coded plots within outlined fields for Mogok data [11,13]. Note extreme Ga/Mg ratios in a proposed "skarn" field for Le-U ruby (red diamonds) and Ohn-bin-ywe-htwet sapphire (mauve diamonds) [13], compared with Ga/Mg ratios plotted for Thurein Taung and Mong Hsu, Myanmar, Barrington Tops, Macquarie River and New England, East Australian ruby suites (Figure 18).

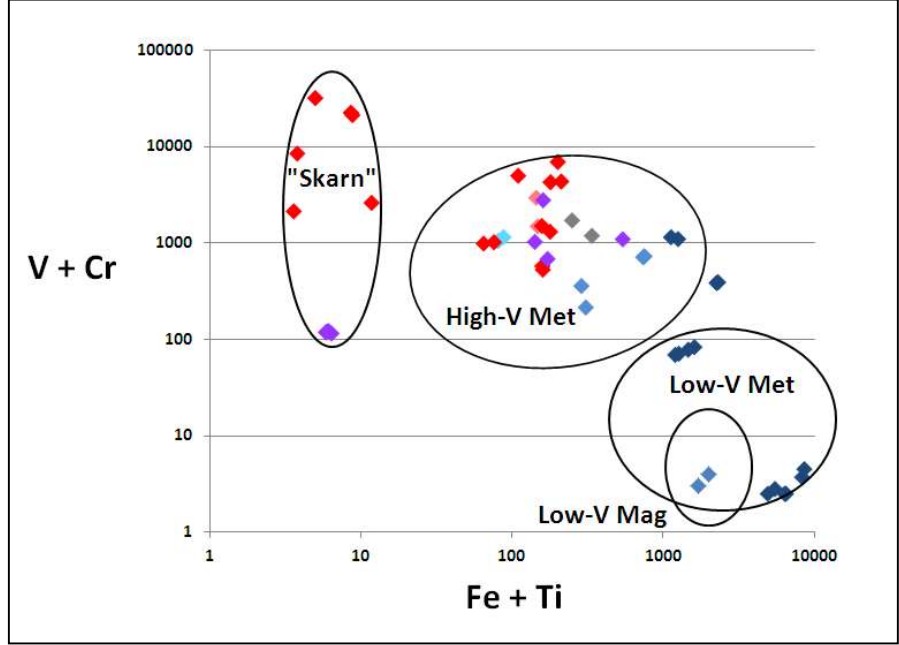

**Figure 21.** V + Cr vs. Fe + Ti diagram showing Mogok gem corundum, as color coded plots, as in Figure 19, within outlined fields for Mogok data from [11,13]. Note the extreme V + Cr enrichment and extreme low Fe + Ti for the suggested Mogok "Skarn field". Such low Fe + Ti is not shared by the other Myanmar or East Australian ruby data in this paper or in the cited literature.

### 4.3. Ruby Diversity and Geographic Typing

This study examined two ruby sites from Myanmar and three ruby sites from East Australia, using rubies derived from quite diverse geological settings. Some ruby suites came from known host rock settings and others from alluvial deposits as xenocrysts transported by basalts that erupted through hidden fold-belt settings. The color zoning and geochemistry of the different ruby suites and their

age formation relationships show considerable diversity within in both inter- and intra-continental settings. The diversity described here is only a limited sample of the potential diversity of ruby deposits world-wide, as revealed in a selected range of major surveys [2,3,15,34,72–76].

Several ruby suites within this intra-continental comparison show unusual characteristics that seem ideal for geographic typing. The Mong Hsu zoned ruby-sapphire stands out, not only for its contrasting dark cores and red rims, but also for added complexity. The zoning also includes blue patches that differ from the dark cores in showing O–H stretching, when examined by FTIR absorption spectrometry [77]. Within some Mogok rubies, very high V levels, exceeding 30,000–50,000 ppm, with V/Cr > 1–26, as at Sin-Khwa, would be distinctive [11]. Furthermore, extreme Si-Ca-Ga-enriched ruby, as at Le-u seems unique [13]. The Le-u ruby was retrieved from an alluvial placer, where rubies were derived from marbles that had interacted with $CO_2$-rich fluids and nearby syenite intrusion [78]. Among East Australian rubies, Barrington Tops, Macquarie River and New England suites all show distinctive features. Barrington Tops ruby composites, although similar in mineralogy and trace element chemistry to West Pailin, Cambodian examples [79] differ from them in their $\delta^{18}$O isotope ranges [9]. Macquarie River ruby shows extreme Mg enrichment relative to Fe and Ti [9] and in a Mg ($\times$100)–Fe–Ti ($\times$10) diagram all plots fell in the end-member Mg apex (Mg 90–100, Fe 0–10, Ti 0–10). New England color-zoned, Ga-enriched ruby, although approached in Ga values elsewhere, e. g., East Africa, differs in Cr-colored red cores trending to non-red rims [48], rather than the reverse trend [80].

A plethora of detailed data now exists on ruby characteristics available from many global sites, as exploration, mining and gem researchers, gem institutions and trade laboratories continue to open-up studies on this valuable gemstone [2,81–83]. This, with improved techniques of characterizing ruby through analytical advances [84], and more sophisticated statistical methods of identifying links/differences in analytical data from known sources [85–88], will enlighten future studies. Detailed mineral assemblage studies and P-T conditions of their formation within ruby-host lithology will become further refined to use as potential exploration vectors for new ruby deposits [5].

## 5. Conclusions

Several ruby deposits in two different tectonic terranes provide diverse ages and chemistry.

Myanmar ruby ages of ~32 Ma at Mogok and at 23 Ma at Mong Hsu reflect the post-Paleozoic collision history and felsic magmatic activity in Myanmar, while older diverse ruby generated within dismembered fold belts in East Australia were delivered to surface placers by post-rift basalts.

Comparisons of ruby trace element chemistry from the Myanmar and Australian suites show considerable geographic distinctions, with carbonate and mafic-ultramafic affinities for Myanmar suites, and metamorphic and metasomatic affinities dominating in Australia.

Both Myanmar and East Australia ruby suites include examples of unusual Ga-rich ruby which appear to indicate generation involving magmatic inputs.

Considering the diversity and chemical distinctions shown by the Myanmar and Australian comparisons, in the context of global ruby distribution and detailed data accumulating on many ruby suites, geographic typing of ruby origins seems to be a viable future proposition.

**Supplementary Materials:** The following are available online at http://www.mdpi.com/2075-163X/9/1/28/s1.

**Author Contributions:** F.L.S. assembled the East Australian gem data, compared it with Mogok, and Mong Hsu data, interpreted the results of the analyses and wrote the script. K.Z. drafted the manuscript, provided Myanmar expertise and funded analytical work. S.M., J.T., K.G. and M.M.Z. provided technical input and ran the LA-ICP-MS and SEM geo-chronological and chemical analytical work. T.T.N. provided geology background, preliminary data on Mong Hsu ruby and additional samples and technical input. K.T. provided samples for analysis and field data on Mogok localities. S.J.H. provided data on New England ruby and plotted Myanmar and Australian diagrams.

**Acknowledgments:** The Editor of *Minerals* and Guest Editors are thanked for organizing the special issue on "Mineralogy and Geochemistry of Gems" and extending an invitation to submit a contribution for consideration. This particularly includes the Australian guest editor, Ian Graham, School of Biological, Earth and Environmental Sciences, University of New South Wales, Sydney. The pre-submission draft manuscript was read by Ross Pogson, Geoscience, The Australian Museum, Sydney. Bruce Wyatt, Port Macquarie, New South Wales, Australia assisted

in tabulation of analytical results. Paula Piilonen, Canadian Museum of Nature, Ottawa, allowed use of an Asian–Australian gem deposit map for modification as Figure 1 in this study. Three reviewers are thanked for constructive comments and editorial staff provided helpful handling of the manuscript for publication.

**Conflicts of Interest:** The authors declare no conflict of interest.

## Appendix A

**Table A1.** LA-ICP-MS trace element values (ppm) [1], Thurein Taung ruby.

| Crystal No. Spot | Mg | Ti | V | Cr | Fe | Ga |
|---|---|---|---|---|---|---|
| **1.**1 (478), rim | 116 | 171 | 112 | 4191 | 60 | 50 |
| **1.**2 (479), core | 94 | 142 | 102 | 3841 | 65 | 48 |
| **2.**1 (480), core | 50 | 200 | 243 | 2883 | 79 | 101 |
| **2.**2 (481), rim | 49 | 329 | 250 | 2785 | 65 | 98 |
| **3.**1 (482), rim | 45 | 63 | 310 | 2306 | 122 | 32 |
| **3.**2 (483), core | 63 | 94 | 386 | 618 | 117 | 39 |
| **4.**1 (484), rim | 27 | 38 | 281 | 2183 | 44 | 155 |
| **4.**2 (485), core | 40 | 64 | 296 | 2340 | 49 | 152 |
| **5.**1 (486), rim | 104 | 1241 | 401 | 2731 | 80 | 20 |
| **5.**2 (487), core | 100 | 203 | 421 | 2917 | 91 | 22 |
| **6.**1 (488), rim | 204 | 339 | 190 | 1073 | 483 | 125 |
| **6.**2 (489), core | 189 | 283 | 185 | 1035 | 461 | 128 |
| **7.**1 (490), rim | 63 | 96 | 145 | 4599 | 38 | 41 |
| **7.**2 (491), core | 77 | 120 | 156 | 4406 | 35 | 43 |
| **8.**1 (492), rim | 69 | 99 | 110 | 432 | 45 | 22 |
| **8.**2 (493), core | 59 | 93 | 125 | 3505 | 24 | 20 |
| **9.**1 (494), rim | 88 | 131 | 386 | 1909 | 353 | 170 |
| **9.**2 (495), core | 35 | 53 | 110 | 902 | 225 | 119 |

[1] Values based on assumed Al content of 52,900 ppm. < = value bdl.

**Table A2.** LA-ICP-MS trace element values (ppm) [1], Mong Hsu ruby sample sets.

| Crystal No. Spot | Mg | Ti | V | Cr | Fe | Ga |
|---|---|---|---|---|---|---|
| **Sample set 1** | | | | | | |
| **1.** rim1 (red) | 27 | 336 | 231 | 3454 | 13 | 75 |
| **1.** core (blue grey) | 76 | 1168 | 254 | 4625 | 14 | 66 |
| **1.** rim2 (red) | 62 | 487 | 238 | 3342 | 32 | 75 |
| **2.** rim1 (red) | 35 | 654 | 628 | 2815 | 10 | 78 |
| **2.** core (red) | 67 | 1397 | 649 | 4470 | <8 | 71 |
| **2.** rim2 (red) | 34 | 580 | 595 | 2780 | 19 | 76 |
| **3.** rim (violet) | 75 | 3201 | 73 | 1393 | 22 | 76 |
| **3.** core (violet) | 42 | 2550 | 75 | 2780 | 17 | 80 |
| **4.** rim1 (light red) | 55 | 1329 | 304 | 446 | 9 | 53 |
| **4.** core (light red) | 95 | 2145 | 316 | 919 | <9 | 48 |
| **4.** rim2 (light red) | 49 | 1137 | 352 | 429 | 15 | 52 |
| **Sample set 2a** | | | | | | |
| **1.** rim (red) | 37 | 383 | 187 | 4476 | 38 | 81 |
| **1.** core (blue black) | 84 | 585 | 658 | 9698 | 54 | 94 |
| **2.** rim (red) | 73 | 1285 | 301 | 5524 | 18 | 81 |
| **2.** core (blue black) | 149 | 2633 | 348 | 6736 | 14 | 85 |
| **3.** rim1 (red) | 147 | 1367 | 1012 | 7771 | 51 | 88 |
| **3.** core (blue black) | 187 | 1358 | 398 | 8180 | 37 | 85 |
| **3.** rim2 (red) | 42 | 616 | 270 | 2033 | 20 | 86 |
| **4.** rim1 (red) | 131 | 2079 | 306 | 2705 | 37 | 78 |
| **4.** core (blue black) | 58 | 889 | 349 | 2011 | 31 | 88 |
| **4.** rim2 (red) | 29 | 324 | 317 | 1961 | 40 | 92 |

**Table A2.** *Cont.*

| Crystal No. Spot | Mg | Ti | V | Cr | Fe | Ga |
|---|---|---|---|---|---|---|
| **Sample set 2b** | | | | | | |
| **1.** rim1 (red), #18 | 29 | 42 | 320 | 16,388 | 31 | 88 |
| **1.** core (blue black), #17 | 41 | 100 | 591 | 27,386 | 33 | 81 |
| **1.** rim2 (red), #19 | 20 | 63 | 269 | 16,013 | 42 | 77 |
| **2.** rim1 (red), #24 | 69 | 1050 | 352 | 2994 | 23 | 75 |
| **2.** core (blue black), #23 | 226 | 2296 | 508 | 5871 | 44 | 85 |
| **2.** rim2 (red), #25 | 146 | 1136 | 465 | 6919 | 34 | 75 |
| **3.** rim1 (red), #27 | 47 | 794 | 240 | 959 | 11 | 92 |
| **3.** core (blue black); #26 | 103 | 681 | 257 | 1194 | 17 | 103 |
| **3.** rim2 (red), #28 | 47 | 77 | 368 | 3195 | 12 | 105 |
| **4.** rim1 (light red), #30 | 24 | 189 | 183 | 4811 | 12 | 68 |
| **4.** core (light red), #29 | 75 | 1136 | 228 | 3226 | 15 | 87 |
| **4.** rim2 (light red), #31. | 104 | 1492 | 195 | 2607 | 34 | 78 |

[1] Values based on assumed Al of 52,300 ppm. < = value bdl. # spot no., in Figure 6.

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
