# Peer review of "Diversity in Ruby Geochemistry and Its Inclusions: Intra- and Inter- Continental Comparisons from Myanmar and Eastern Australia"

_minerals, doi:10.3390/min9010028_

Reviewer 1 Report

Review of manuscript “Diversity in Ruby Geochemistry and its Inclusions: Intra - and Inter - Continental Comparisons from Myanmar and Eastern Australia” by Frederick L. Sutherland, Khin Zaw, Sebastien Meffre, Jay Thompson, Karsten Goemann, Kyaw Thu, Than Than Nu, Mazlinfalina Mohd Zin, Stephen Harris

Dear authors,

First of all, it was a pleasure to me reviewing your interesting manuscript dealing with important ruby deposits in Myanmar. The manuscript contains well-performed LA-ICP-MS trace-element and U-Pb analytical data, the last was likely for the first time applied to titanite and ilmenite inclusions within ruby host. Results on ruby trace-element chemistry from primary Myanmar and secondary Australian ruby and sapphire occurrences included to binary discriminant diagrams might be potentially used further for tracing the ruby origin. Therefore, in my opinion, the manuscript is in an acceptable format for publication in Minerals journal. I recommend it acceptance after a MINOR REVISION.

MINOR COMMENTS:

 1.  The manuscript is lack of citing the literature published previously on LA-ICP-MS U-Pb in situ dating of different inclusions in corundum applied for their origin identifications. I recommend citing some of these articles:

 Klemens, L. (2015) The Journal of Gemmology. 34(8), 692–700

Sorokina, E.S. et al. (2017) Mineralium Deposita. 52 (5), 641–649;

Saul, J.M. (2017) International Geology Review. 1–22. DOI:10.1080/00206814.2017.1354730.

2.  I would also cite Muhlmeister et al. paper as they for the first time applied the chemical method for geological typing of gem corundum deposits. Muhlmeister et al. (1998) G&G. 34(2), 80–101.

3.   Line 50. I am not sure about Archean age of ruby, otherwise, please cite the appropriate references in the text. As far as I know, the oldest age of ruby deposits is Neoproterozoic age:

 Saul, J.M. (2014) A Geologist Speculates. Paris, Les 3 Colonnes, 159 p.;

Giuliani, G. et al. (2014) Chapter 2. In: Mineralogical Association of Canada Short Course 44, Tucson AZ, February 2014, 29–112;

 4.   Line 54–55. Since the ruby is expensive gemstones and not frequently applied even for micro-destructive technique, I would also add the studying of solid/melt inclusions for origin determination as well (as most of them are common for particular ruby deposits) and the optical microscopy combined with Raman spectroscopy as primary non-destructive techniques for their testing. Appropriate references for citing:

 Gubelin, E.J., Koivula, J.I (2008) Photoatlas of inclusions in gemstones. Vol.3. Opinio Verlag, Basel, 672 p.

Hughes, R.W. (2017). Ruby & sapphire: A gemologist‘s guide. RWL Publishing/Lotus publishing, Bangkok. 816 p.

5.  Please provide the Concordia age obtained on zircon, baddeleyite, titanite, and ilmenite references used for LA-ICP-MS U-Pb dating in the Methods section.

6.   Please indicate the core and rim measurements in the Table A.1. I am wondering, which trace-elements finally caused the color zonation in Mong Hsu rubies? Please indicate it in the text.

7.   I would exclude the Si and Ca in the Tables with ruby trace-elements measured by LA-ICP-MS and their mentioning in the text as they give interference by LA-ICP-MS. Please have a look the following reference in this regard:

 Stone-Sundberg, J. et al. (2017) G& G, 53(4), 438–451.

8.    General comment on LA-ICP-MS U-Pb part: please provide (if possible) the cathodoluminescense images for all measured zircon and baddeylite grains with the spots locations and ages on them, as well as BSE images for ilmenite and titanite.

9.   The elder zircon and baddeleyite U-Pb ages could be explained by the different closure temperature comparing to titanite and ilmenite. Zircon (and baddeleyte) common dating the peak metamorphism (Pb diffusion in zircon structure occures at about 900–1000 °C) whereas the rest of minerals (titanite, ilmenite) with much lower closure temperatures signifing only the coolling ages after peak metamorphism (please, have a look our paper in Mineralium Deposita, 52 (5), 641–649). Thus, I would not exclude 2 zircon U-Pb measurements in Thurein Tang deposit constraining likely the metamorphic peak in the area at about 40 – 50 Ma, which is correlated with Himalayn orogeny. Besides, you may use the zircon and titanite dating for P-T path. The same is for Mong Hsu.

10.   Table 3. The name of inclusion should be nepheline (not nephelinite). Si values are very high, if it in oxide form in the Table it should be virtually recalculated to element as the rest of measurements were provided in elements.

11.    Figure 3 requires scale and north arrow.

12.    I would indicate the deposits at a Figures 17 and 18.

13.   Please check the citing of all the figures in the text, most of them is mixed up.

14.   Please indicate the Mong Hsu ruby deposit at a Figure 4.

08/12/2018

Author Response

REVIEWER 1 : Response to points made on Ms Minerals -406108

       The reviewer made 14 points that were considered needing addressing.

       Points 1–4. The ms needed extra citing of literature previously dealing with in situ dating of different inclusions in corundum for origin identification and also added literature for non-destructive means Eight references were outlined for citing and full referencing in the Introduction section.

Point 4. The Archaen age for ruby formation was questioned and if so need a supporting citation. This was provided by citing last reference in the script (77) to a new initial position [3] directly in context with the Archaen claim.

All the above points 1–4, suggesting 9 new citations/ referencing with appropriate new context, have been inserted into an expanded new introduction 1.1 Background, lines 52-64 (FLS).

       Point 5. Asked for the Concordia age provided on zircon, baddelyite, titanate and ilmenite references used for LA-ICP-Ms U-Pb dating in the Methods section. Response to this is copied here from co-author J.T.   These reference material U-Pb ages are reported in the Geochronology appendix that is an Excel file in the supplementary material. Does the reviewer wish to have an additional table in-text with the reference material results? This can easily be done, just was a little unclear what was requested from the comment. No reference materials are available for ilmenite, it was analysed as a common Pb anchor. This is the Table that can be supplied as an in- text Table, Appendix Table or Supplementary Table, if needed. 

Table X. Geochronology reference materials treated as unknowns compared to published values:

Material

LA-ICPMS   Age (Ma)

Published   age (Ma)

Fish   Canyon Tuff Titanite

26.8±2.9

28.13±0.48

100606   Titanite

429.3±7.3

432.02±0.64

Plesovice   Zircon

339.2±3.7

337.13±0.37

        Point 6. Please indicate the core and rim measurements in Table A1, to examine trace elements that may cause color variation in Mong Hsu rubies and indicate in text.  

The core and rim designations for analyses in Table A1 were added. This caused minor adjustment to be made on a few figures in Table 1, summary results, which were adjusted, Lines 507-508. Text has been added re comparisons of Table 1 (Thurein Taung) and Table 2 (Mong Hsu) in relation to Mong Hsu color-causing zonation, Lines 509-519 (FLS).  

       Point 7. Suggests exclude Si and Ca from Tables A1, A2, on grounds of interference effects in measuring LA-ICP-MS results of these elements and modify text citing a reference Stone- Sunderberg et al. 2017 in this context. This has been done for Tables A1and A2 (Lines1101–1123), with the reference cited as [60] and listed in References (lines 1370–13711). A further explanation is copied from a co-author (JT). We normally measure 29Si and 44Ca, which are interference free, however in one of the sessions we accidently measured 28Si. Agreed that there can be an interference on 28 from 27Al+1H and on 43Ca from 27Al+16O, however these elements were monitored to assess inclusions of other mineral phases during the ablation and not used in a quantitative sense to assess the trace element composition of the corundum.      

        Point 8. Provide if possible cathode luminescence for measured zircon and baddeleyite grains, with spot locations and ages, and BSE images for ilmenite and titanate. Response to this is copied from co-author KT, with reasons why these were not made or considered worthwhile within the context.

Generally, the size of the zircon roughly the same diameter as the laser pit (see inclusion 3 in Figure 12) so no CL imaging was done as we wouldn’t have had the option of avoiding any potential cores. Given that the CL response is mostly from defects in the crystal lattice from trace element substitution, we didn’t think CL imaging was useful for these particular zircons. BSE of the zircon and Titanite was done and representative images are in the text.

The ilmenite were analysed for an estimate of common Pb composition of the inclusions since the zircon analyses in Figure 9 were discordant due to common Pb. While we could have used model Pb from Stacey and Kramers (1975) paper, we thought it better to have an independent estimate of common Pb for the correction.    

        Point 9. Provide a potential explanation based on different closure temperatures in zircon in comparisons between titanate and ilmenite, which have lower closures temperature. This may mean high T resetting of zircon at peak metamorphism and later lower cooling in titanate and ilmenite .This may correlate with at 40-50 Ma during the Himalayan Orogeny. Need to consider a paper in Mineralium Deposita 52(5), 641-649. This can provide a PT path. The same also for Mong Hsu.

Comments on this as an unsure procedure, based on analytical results in the paper are given by JT.

The three zircon analyses from the Thurein Tang deposit show significant variation in age (90 to 50Ma), well outside analytical uncertainty of each other and the former of which is well prior to the Himalyan orogeny. This is why we did not interpret these further. If we had found more zircon inclusions in the rubies, potentially we could have said something more meaningful.                                                                                                       

Point 10. Table 3. Mineral name nephelinite should be nepheline, not nephelinite. Si over-high oxide value needs recalculation to Si wt %. Adjusted, FLS with new Table No.2, Lines 748-749, with recalculated Si value and other minor improvements by co-author MG.

Point 11. Fig 3 needs north direction and scale. This was added as north direction and distance scale given in new Figure map caption and scale and V/H added for cross section A-B, now new Figure No. 4, Lines 2215-220, FLS.

Point 12. The deposit indications in Figures, in Figs 17, 18 need indication.

This is relevant to the plots in the skarn field and have been given color –coded designation in the Figure Caption, now new Figure Nos 19, 20, Lines 959-965 and 1014-1020, FLS.

Point 13. Check citing of all figures in the text to remove mix ups.  Done, FLS.

Point 14.Please indicate Mong Hsu ruby deposit location in Figure 4.

This is indicated for map and cross section in re-configured Figure 4, caption, Lines 210–221.

Reviewer 2 Report

Dear editor and authors

 I have now read manuscript "minerals-406148" titled: Diversity in Ruby Geochemistry and its Inclusions: Intra - and Inter - Continental Comparisons from Myanmar and Eastern Australia.

 This study presents new trace element data and U-Pb ages for rubies from Myanmar and Australia, and adds to a growing database, which can be used for fingerprinting of ruby provenance globally.

This is a welcomed contribution and I would recommend accepting this work after minor revision.

My only criticism is the quality of Figure 2, which uses a range of very similar beige colors that makes it impossible to separate the different lithological units. Figures 14 and 15 also needs to be improved in terms of resolution and text size.

Below I give a list of specific suggestions for improving the manuscript, which I hope will be of use for the revisions.

Best regards,

Specific comments

Line 21: Use "petrogenetic" instead.

L22: Rock-hosted.

L 29: Please add that the trace element data are done by in situ LA-ICP-MS.

L52: Transportation in singular.

L79: Add ages of intrusive rocks.

L80: Please elaborate on the nature of the ultramafic rocks. Are they intrusive or part of the early high-grade rocks. They could be important for buffering the silica activity.

L133: fold belt "in" Australia.

L137–138: This sentence is unclear. Granite from mantle? Intruded down? Please re-write this part to avoid any confusion.

L251: "homogeneous".

L252–254: What are the values reported as negative? Please check if these number are correct.

L403: Use a different word than "pulsating", because this is time-scale dependent.

L515: Do you really mean "script" as in a computer program or do you mean manuscript?

Author Response

REVIEWER 2:   Response to points made on Ms Minerals-406148.

Two matter were raised generally. Similar colors in the Mogok geologiy Map, Figure 2, made it hard to distinguish separate lithological units.

While the colors are subdued, on raising the matter with co-authors most though there was less problem where most gem site are disposed across the two main marble units, which included the gem site studied in the submission.

The authors had two other alternative maps available for use. One was a more strongly colored lithology published map, but this had a serious flaw in its overlay of deposit sites that incorrectly displaced them from the host marbles into the incorrect gneiss lithology. The gem site numbering also became incorrect in positioning. The deposit numbers are also not needed, as the single ruby site studied in the submission is not identified on this map.

The other available map, published in the Geological Society of London Myanmar Geology Volume, Chapter 23, is highly colored, but garish and cluttered with numbered/named deposits that make lithological /deposit relations hard to follow and again are unneeded detail for a single deposit focus in the submission.   

The present map used in the submission clearly shows the disposition of the gem across the geology units and is the much preferred choice for this study. The exact Thurein Taung site is given relative to the map scale in the new Figure 2 caption, Lines 118-121 and by the precise long., lat grid location in in the following text, Line122.

The other matter required greater size and resolution for Figures 14, 15. This has been implemented, at resolution of 300 dpi in Fig. 14, with the triple panel Fig. 15, being broken up and replaced with larger Figures with better resolution, as Figs 15, 16, Lines 891-895, Fig 17.  925-929. 17,  SH, LS.

Specific points.

L 21. Use petrogenetic instead. Done.

L 22. ‘Rock-hosted’. Done.

L 79.  Add ages of intrusive rock. Done. Now L 82–89.

 L 80. Elaborate nature off the ultramafic rocks. Are they intrusive or part of early high grade rocks, to judge buffering of silica activity? Done. Now L 84–86.

L 133. Fold belt ‘in’ Australia. Done.   ‘in’ added, new L 283.

L 137–138. Unclear sentence. Rewrite ‘granite from the mantle’. ‘intruded down’?

Rewritten with greater elaboration, now L 290–291., with extra citation [45], and its reference, New L 1287–12880. New information on ultramafic slices with extra citation [46], new Lines 291–293 and reference listing, L 1289–1291.

L 251. Word ’homogeneous’. Replaced by ‘homogenous’. New L 517.

L 252–254. What are values reported as negative? Please check if numbers are correct. This section clarified and numbers altered where needed. New L 515-519.

L 403. Use different word to ‘pulsating’ a time-scale dependant. Replaced by ‘multiple’, New L 878.L 515. Is script meant as computer script or manuscript? Replaced by ‘ms’. New L 1095.

Reviewer 3 Report

Overall the manuscript present a nice study with a large amount of data. there are some issue on the presentation style for which the manuscript seems to have been rushed a bit. perhaps you can see the input of several authors and there are inconsistency throughout the paper.  See below and on the file for more detailed comments.

All the figures and graphs need to be improved as suggested in the text. The quality of some is particularly poor and not well presented mostly.

Figures 5 to 9 are not recalled in the main body text at all!

Tables numbers are incorrect! Table 1 and 3 but no table 2

Author Response

REVIEWER 3: response to comments on Minerals Ms -406148.

OVERALL

All the Figures and graphs need to be improved as suggested in the text. Figures 5-9 are not recalled in the main body of the text at all.

New text is inserted into the script which brings in comparisons of the materials shown in the Figures, New L 302–308, which include Fig. Nos 5; 6a,b; 7; 8; 9a,b; 10.

POINTS IN TEXT.

Line 62. Last sentence not needed, should be in References and Notes.

Lines 62-64 deleted. Citation L [22] moved to Line 65 at start of previous sentence and reference listed at new L 1184–1187, FLS.

Figure 1 Map. Not good enough, needs redrawing and starting from scratch.

 This Map has been deleted and replaced by a similar, but new higher resolution map, with a new citations in the Figure caption [24] acknowledging use of new map. The previous gem site nos have been deleted for simplicity and the Myanmar and East Australian localities studied in the submission are now indicated by arrows inside the map border.

Line 74–75. Queries wording in sentence. Reworded in script, New L 79, FLS.

Figure 3a, b, Lines 90-98. Figures are pixelated, need higher resolution. Rock symbols, 3b, need explanation in Fig. caption.  

Images improved to 300 dpi, KT. Rock symbol information put in Fig. caption, New L 1868–177, FLS.

Figure 4, Lines 111-114. Rearrange cross section below geomap and enlarge. Explanation should be Legend. Change dip/strike.

Map is enlarged with cross section underneath. Explanation column is now dealt within the Figure caption, New L 211-221, FLS.

Figure 6 a,b, c, Lines 153-159. Scale not visible, nor labels on spots. Titanite label in (c) not needed.

New better resolution images, with scales, numbered spots are now included, New L 338–343, JT.

Image 6c is now separated from 6a, b, for better viewing. The titanite label (c) was left in place, as it covers previous label as zircon, an error in initial visual ID, New L 344–352, FLS.

Figure 7, Lines 159-161. Very confusing Figure, with many info needing insertion into Figure caption.

The sample core image is now detached from the surrounding info, which is dealt with by citation in the caption. New Figure no. 8, new L358–363, FLS.

Figure 8, Lines 162–165. A and b too large in pics.

Have been unable to deal with this, although have tried, without Photo Shop or original images. Perhaps Editorial expertise at Minerals may be able to manage this? New Figure 9.

Figure 9. Lines 166–168. Is a better image of the stone without a partial cut off available?

 Although great part of stone is captured in the image, a complete view is not available. The stone was taken at this angle to show color depth and fine-scale zoning. New Figure No.10, L 401–403, FLS.

Lines 244-249, below Table 1. All u.m. needed every time for max and av values.

These have been entered where needed, New L 509–514, FLS.

Figure 11, Lines 299–302. Why is the graph shifted to the right?

This was a box problem and has been rectified. New L 714–718, JT.

Line 329. Table heading. Should be Table 2. Renumbered, New L 748, FLS.

Figure 14. Lines 385–389. Low resolution image. Upgraded to 300 dpi., New L 859–864, FLS.

Figure 15. Lines 407–409. Too small, needs re-arranging so all graphs bigger and visible with writing.

The triple panel in Figure is broken into separate Figures each with better resolution and visibility. New Figs15, 16, L891–895, L 903–906, New Fig. 17, L 925–929. SH, FLS.

Line 513. Supplementary Material Table S1: Geochronology. Where is this?

 This was supplied separately with original submission to the Editor for forwarding on to Reviewers.  The Table will be also submitted with the revised ms.